# Pharmacological Intervention with 4-Phenylbutyrate Ameliorates TiAl6V4 Nanoparticles-Induced Inflammatory Osteolysis by Promoting Macrophage Apoptosis

**DOI:** 10.3390/bioengineering12070701

**Published:** 2025-06-27

**Authors:** Guoyin Liu, Haiyang Gong, Tianting Bai, Yahui Fu, Xin Li, Junhao Lu, Jianning Zhao, Jianmin Chen

**Affiliations:** 1Department of Orthopedics, Jinling Hospital, Affiliated Hospital of Medical School, Nanjing University, Nanjing 210093, China; liuguoyin0425@163.com (G.L.); m15951922326_1@163.com (T.B.); 2Department of Orthopedics, Jinling Clinical Medical College, Nanjing University of Chinese Medicine, Nanjing 210002, China; 3Department of Rehabilitation, Jinling Hospital, Affiliated Hospital of Medical School, Nanjing University, Nanjing 210093, China; rehabi1314@163.com; 4Medical School, Nanjing University, Nanjing 210001, China; fuyahui1126@163.com; 5Department of Orthopedics, Central Military Commission Joint Logistics Support Force 904th Hospital, Wuxi 214044, China; lixin198611@126.com; 6Department of Orthopedics, Nanjing Hospital of Chinese Medicine, Affiliated Hospital of Nanjing University of Chinese Medicine, Nanjing 210022, China; haljh2005@sina.com

**Keywords:** osteolysis, aseptic loosening, wear particles, 4-phenylbutyrate, apoptosis, inflammation

## Abstract

Macrophage apoptosis, along with inflammation in the interface membrane, has been demonstrated to be significant in the pathogenesis and development of particle-induced periprosthetic osteolysis and aseptic loosening. Additionally, the apoptosis of macrophages is considered an indicator of the resolution phase of inflammation and the transition to normal tissue healing. Therefore, targeting macrophages presents a promising strategy for both the prevention and therapeutic management of periprosthetic osteolysis. In this study, we explored the therapeutic potential of chemical chaperone 4-phenylbutyrate (4-PBA) as a pharmacological intervention aimed at modulating macrophage behaviors, particularly focusing on the processes of apoptosis, inflammation, and osteoclastogenesis in a murine model of TiAl6V4 nanoparticle (TiNP)-induced osteolysis. The results derived from in vivo studies conducted on the murine model provide compelling evidence that TiNPs could trigger osteolysis, activate inflammatory cell infiltration, and promote the differentiation of osteoclasts, accompanied by a notable rise in apoptosis at the osteolytic interface periosteum. The severity of TiNP-induced osteolysis, chaotic bone morphology, extensive bone erosion and destruction, occurrence of infiltrating inflammatory cells, and quantity of osteoclasts were attenuated following co-intervention with 4-PBA. Furthermore, the levels of apoptosis, in conjunction with apoptosis-regulated proteins Bcl-2 and Bax, were accentuated following 4-PBA co-intervention, indicating that the TiNP-induced osteolytic interface periosteum environment exhibited a greater propensity for apoptosis due to the pharmacological intervention of 4-PBA. Notably, the use of 4-PBA as a standalone treatment demonstrated comparatively low levels of toxicity and was deemed to be experimentally safe in mice. These findings indicated that 4-PBA may ameliorate the severity of particle-induced osteolysis by inhibiting the inflammatory response and promoting macrophage apoptosis in a manner that may be beneficial for therapeutic strategies. Thus, pharmacological intervention with 4-PBA appears to be a viable option for addressing osteolysis and aseptic loosening resulting from exposure to wear particles, combining efficacy in promoting apoptosis with a favorable safety profile.

## 1. Introduction

Joint replacement, which involves the implantation of permanent prosthetic components, is considered one of the most groundbreaking elective surgical interventions designed to enhance the mobility of individuals suffering from dysfunctional joints [1]. However, a significant challenge associated with joint replacement is the phenomenon of aseptic loosening, which is often a consequence of periprosthetic osteolysis [2]. Addressing this issue is of paramount importance for enhancing the long-term success and durability of joint replacement procedures, particularly in relatively younger patients who may require these interventions to endure for an extended period [3].

Periprosthetic osteolysis is fundamentally linked to the interaction between wear particles and host cells [4,5]. This interaction leads to the phagocytosis of particles, which activates various immune and tissue cells, including macrophages, fibroblasts, lymphocytes, and osteoclasts [6]. These activated cells secrete pro-inflammatory cytokines (e.g., TNF-α, IL-1β, and IL-6) [7] and osteoclastogenic molecules (e.g., RANKL and M-CSF), which facilitate osteoclast formation and activation [8]. Although TRAP is not a secreted factor, its expression in osteoclasts and osteoclast precursors during differentiation is essential for their function in bone resorption [8]. The production of reactive oxygen species (ROS) and proteolytic enzymes also occurs, further compounding the osteolytic response [9]. The osteolytic response is amplified by these factors via various biological functions, including the differentiation of macrophages into osteoclasts, which are responsible for the process of osteolysis through the lacunar bone resorption mechanism. Additionally, these factors contribute to the induction of apoptosis in adjacent cells, further exacerbating local tissue damage. Moreover, they facilitate the release of further cytokines, which serve to sustain and amplify the ongoing cycle of inflammation and bone resorption [10]. Evidence has accumulated to show that macrophages are the key components driving bone destruction in periprosthetic osteolysis [11]. These cells play a major role in inflammation and the formation of osteoclasts, and they have been identified as the primary pathophysiological mechanism associated with particle-induced osteolysis (PIO) [12]. The presence of apoptotic macrophages has been observed in the osteolytic interface membrane [13]. Researchers have noted that events related to macrophage apoptosis are associated with periprosthetic osteolysis, potentially providing specific targets for therapeutic modulation [13,14,15,16,17,18].

Non-biologic immunomodulators, particularly bisphosphonates (e.g., pamidronate, risedronate, alendronate, ibandronate, clodronate, and zoledronic acid), have shown potential in managing periprosthetic osteolysis [19,20,21,22]. Bisphosphonates are widely used for osteoclast-driven bone diseases, like osteoporosis and Paget’s disease. They have historically been employed to theoretically reduce revision rates and extend the clinical survivorship of joint replacements [20,21]. Their ability to mitigate bone resorption by modulating osteoclast activity and facilitating macrophage apoptosis reinforces the idea that inducing macrophage apoptosis could be a viable therapeutic strategy for various bone-related conditions [19]. Consequently, addressing osteolysis via promoting macrophage apoptosis represents a promising and worthwhile approach in clinical therapy.

Endoplasmic reticulum stress (ERS) results from the accumulation of improperly folded or unfolded proteins within the cell and is crucial for cellular homeostasis [23,24]. Cells respond to ERS by initiating the unfolded protein response (UPR) [25]. During the process of PIO, a significant quantity of active proteins is synthesized [26]. This surge in protein production is essential for the various biological responses activated during osteolysis [26]. However, when there is a disruption in the protein folding mechanism, it can result in an imbalance that prompts macrophages to initiate the UPR [7]. The close relationship between UPR and macrophage activation highlights the importance of ERS in PIO mechanisms [27], and its contributions are being increasingly explored for developing strategies to combat implant-associated osteolysis [7,17,28].

Chemical chaperones are reagents that attenuate ERS by enhancing the capacity for protein folding and inhibiting the activation of UPR within ER, with 4-phenylbutyrate (4-PBA) serving as a prominent example [29]. 4-PBA has demonstrated therapeutic potential across various physiological and pathological processes, including inflammation, infection [29], oncology [30], cystic fibrosis [31], type 2 diabetes mellitus [32], neurodegenerative diseases (e.g., amyotrophic lateral sclerosis [33], Huntington’s disease [34], Alzheimer’s disease [35], Parkinson’s disease [36]), degenerative musculoskeletal diseases [37], and autoimmune disorders [38]. This broad applicability underscores its promise as a versatile therapeutic agent.

Recent research has increasingly pointed to the role of ERS and UPR as significant factors in the development of PIO [28]. Furthermore, there is a compelling link between ERS and the broader phenomenon of inflammatory joint destruction, which encompasses such processes as bone erosion and resorption [26]. This connection underscores the complex mechanisms involved in the pathogenesis of such conditions, suggesting that ERS may initiate a cascade of inflammatory responses that contribute to periprosthetic osteolysis. The activation of ERS was documented in models of PIO, and the reduction in ERS with 4-PBA ameliorated the severity of PIO across different inflammatory conditions [7,8,17,26,27,39,40]. 4-PBA exerts potent pharmacological effects through its ability to inhibit inflammation mediated by ERS in models of PIO, as well as in vitro interventions with macrophages [7,8,17,26,27,39,40]. Specifically, research has provided evidence that 4-PBA can induce cell cycle arrest and apoptosis in various cell types through alternative apoptosis pathways, excluding those mediated by ERS [30,41]. Notably, these pathways may include the Jun-N terminal kinase (JNK) pathway [42], disruptions in mitochondrial metabolism [43], or the activation of death receptors signaling [44]. However, it remains unclear whether 4-PBA has the capacity to attenuate the severity of PIO by promoting the process of apoptosis in macrophages or osteolytic interface periosteum, which is predominantly composed of macrophages. Further investigation is needed to clarify the potential mechanism by which 4-PBA may influence these specific cellular responses and its overall impact on the progression of PIO.

While investigations into the impact of 4-PBA on various cellular processes are emerging, the specific mechanisms through which it may influence macrophage activity and, subsequently, inflammatory osteolysis require further exploration [7,8,17,26,27,39,40]. Understanding whether 4-PBA can effectively induce macrophage apoptosis could be crucial in determining its therapeutic efficacy in managing conditions characterized by inflammatory osteolysis. Consequently, it is essential to conduct further research focused on clarifying the connection between the co-intervention of 4-PBA and the functioning of macrophages exposed to particles. TiAl6V4 nanoparticles (TiNPs)-induced osteolysis serves as the predominant model for assessing the impact of pharmaceuticals on PIO. In the present study, we directed our research efforts towards deciphering the efficacy of pharmacological intervention with 4-PBA in ameliorating inflammatory osteolysis induced by TiNPs and its potential link to macrophage apoptosis. Our study shall be an attempt to unveil the missing links through inhibition of ERS in PIO, which still remains unexplored regarding its subsequent involvement in enhancing apoptosis. This investigation will provide valuable insights into how 4-PBA may influence macrophage activities in the presence of these particles, potentially revealing critical mechanisms at play in this biological context. By delving deeper into this relationship, researchers can contribute to the broader knowledge surrounding macrophage behavior and the implications of 4-PBA as a co-intervention agent.

## 2. Material and Methods

### 2.1. Ethics Statement

All animal research protocols were approved by the Ethics Committee of Animal Experiments at Jinling Hospital (Permit No. 2021DZGKJDWLS-00152) and strictly adhered to the Institutional Guidelines for the Care and Use of Laboratory Animals established by Nanjing University.

### 2.2. Nanoparticle Preparation

TiNPs, which were supplied by Dr. Zhenzhong Zhang (College of Materials Science and Engineering, Nanjing University of Technology), exhibited an average particle diameter of 51.7 nm, as measured through transmission electron microscopy [11,26,27,39]. TiNPs were processed meticulously under sterile conditions within a cell culture hood to ensure their purity and to avoid contamination. Different concentrations of solutions were dispersed in the medium for 10 min with a Shumei KQ218 Ultrasonic Cleaner (Kunshan Ultrasonic Instruments Co., Ltd., Kunshan, China). In addition, all particles were confirmed to be free from endotoxins, as assessed using a quantitative limulus amebocyte lysate (LAL) assay (Charles River, Grand Island, UK). This assay achieved a detection threshold of 0.25% EU/mL, ensuring the reliability and safety of the particles for further applications.

### 2.3. Murine Calvaria Resorption Model

A widely recognized and extensively utilized model for investigating PIO is the murine calvaria resorption model [26]. In this model, wear particles and other stimuli are placed onto the calvaria, allowing for direct measurement of their effects on bone. At the onset of the experiment, all mice utilized in the study weighed between 20 and 25 g (Experimental Animal Center of Nanjing University, Nanjing, China). Sixty female C57BL/J6 mice, aged 12 to 14 weeks, were randomly assigned to the following four different experimental groups, each consisting of fifteen mice: the Control group (sham surgery controls), the TiNPs group (TiNP intervention), the 4-PBA group (4-PBA intervention), and the TiNPs + 4-PBA group (combined treatment with TiNPs and 4-PBA). To ensure proper treatment administration, 40 µL of TiNP suspension at a concentration of 30 mg/mL were evenly distributed on the intact periosteum. Additionally, 4-PBA was administered via intraperitoneal injection at a dosage of 300 mg/kg, with treatment occurring on days 0, 1, 3, 5, and 9 following the surgical procedure. No clinically used anti-osteolytic agent, such as bisphosphonates, was included as a positive control group in this study, aligning with the current challenge in the field, which is lacking FDA-approved pharmacological therapies for PIO.

### 2.4. Specimen Retrieval and Histological Processing

Two weeks following successful molding, animals were sacrificed using a carbon dioxide chamber. During the procedure, calvarial caps along with the osteolytic interface periosteum tissues were carefully harvested. Of the collected calvarial caps, two-thirds were preserved in pre-chilled PBS for subsequent toluidine blue (TB) staining and tartrate-resistant acid phosphatase (TRAP) staining. The remaining calvarial caps were allocated for hematoxylin and eosin (HE) staining. Additionally, osteolytic interface periosteum tissues were meticulously collected and weighed using an analytical balance. A portion, constituting one-third, was processed by grinding in liquid nitrogen and subsequently lysed in RIPA lysis buffer (Beyotimme, Nantong, China), supplemented with protein inhibitor cocktails (Sigma-Aldrich, St. Louis, MO, USA). The resulting supernatants were then stored at −80 °C for subsequent Western blotting and quantification of ROS levels, as well as caspase-3 and TRAP activities. Another one-third were subjected to homogenization with a high-speed blender in a medium containing the requisite RIPA lysis buffer and protein inhibitor cocktails, with the final supernatants also preserved at −80 °C for enzyme-linked immunosorbent assay (ELISA) analysis. The remaining one-third were fixed in 4% paraformaldehyde for 24 h, and the resulting paraffin-embedded tissues underwent processing for immunohistochemistry.

### 2.5. The Histopathologic Changes of Osteolysis

The histopathologic changes of osteolysis were evaluated by TB, HE, and TRAP staining (Jiancheng Biotech, Nanjing, China). Bone resorption experiments were conducted on fresh calvarial caps utilizing the TB staining method. Briefly, calvarial caps, whose surfaces were removed carefully, were placed in 0.25% trypsin for 15 min and then left overnight in 0.25 min ammonium hydroxide before being stained in 0.25% TB for 15 min. The calvarial caps were manicured and mounted on slides with buffering glycerin after being washed vigorously and air dried. The extent of lacunar resorption was assessed using a light microscope (Nikon TE2000U, Nikon, Tokyo, Japan) by counting the number of resorption pits. Histomorphometric measurements were performed using Image-Pro Plus analysis software version 6.0 (Media Cybernetics, Silver Spring, MD, USA). The ratio of bone resorption was evaluated by measuring the area of resorption in relation to that of its surrounding box, using microscopic fields at a 40× magnification, and the lacunar resorption ratio was calculated automatically. Similarly, osteoclasts present in fresh calvarial caps were identified via the TRAP staining method.

The inflammatory experiments were conducted on osteolytic calvarial caps using the HE staining method. Briefly, the paraformaldehyde-fixed and paraffin-embedded sections from osteolytic calvarial caps were cut to a thickness of 4 mm and stained with HE solutions (Shanghai Rainbow Medical Reagent Research Co. Ltd., Shanghai, China) for standard histological analysis. Four separate sections per sample were analyzed in an unbiased fashion. The positive cells for the targeted protein within the immunohistological sections were quantified using Image-Pro Plus analysis software 6.0 (Media Cybernetics, Silver Spring, MD, USA).

### 2.6. TRAP Activity and Caspase-3 Activity

The osteolytic interface periosteum tissues were prepared for analysis of caspase-3 enzyme activity (caspase-3 colorimetric assays from BioSource, Liège, Belgium) or for assessing TRAP activity (TRAP Quantification Kit from Jiancheng Biotech, Nanjing, China). All steps of the procedures adhered to the guidelines provided by the manufacturer. Measurements were obtained using a microplate reader set at 405 nm, with variations in absorbance rates being directly proportional to the activities of TRAP and caspase-3. The final dates were normalized based on the total cell protein quantified with a commercial kit, and the results were recorded as U/gprot.

### 2.7. Inflammatory and Osteoclastogenic Cytokine

The levels of inflammatory cytokines (TNF-α, IL-1β, and IL-6) in osteolytic interface periosteum tissues, along with osteoclastogenic cytokines (RANKL and M-CSF) in the same periosteum tissues, were measured using ELISA kits (Jonin Biotech, Shanghai, China). All procedures were conducted following the guidelines provided by the manufacturer. Total protein levels served as an internal control during the analysis.

### 2.8. ROS Content in Osteolytic Interface Periosteum

The content of ROS in osteolytic interface periosteum tissues were measured using the ROS-specific fluorescent probe 2-, 7-dichlorofluorescin diacetate (DCFH-DA; Beyotime, Haimen, China). All steps were conducted following the guidelines provided by the manufacturer. Measurements were taken at an excitation wavelength of 488 nm and an emission wavelength of 525 nm. The variations in absorbance rates were directly correlated with the ROS content. Total protein levels served as an internal control during the analysis.

### 2.9. Western Blotting (WB)

Equal amounts of total protein from osteolytic interface periosteum tissues were separated by 10% sodium dodecyl sulfate–polyacrylamide gel electrophoresis (SDS-PAGE) (1.5 min TRIS·Hcl, pH = 8.8, 30% acrylamide, 10% SDS, AP, TEMED) and then transferred onto nitrocellulose membranes (Millipore, Billerica, MA, USA). To prevent nonspecific binding, nonfat milk was used to block the membranes, after which the membranes were incubated and probed with rabbit polyclonal antibodies targeting Bcl-2 and Bax (Proteintech Group, Wuhan, China), and caspase-3 (Abway Antibody Technology, Shanghai, China). Subsequently, the secondary antibody horseradish peroxidase (HRP)-conjugated anti-rabbit IgG (Servicebio Technology, Wuhan, China) was then applied. The purified protein’s molecular weight was determined by SDS-PAGE, and the band was excised for protein identification using a chemiluminescence detection system (Shenhua Science Technology, Hangzhou, China). Band density was analyzed with ImageJ 1.41 (National Institutes of Health, Bethesda, MD, USA).

### 2.10. Detection of Apoptosis Through TUNEL Staining

The paraformaldehyde-fixed and paraffin-embedded osteolytic interface periosteum tissues were cut to 4-um-thick longitudinal sections, mounted on precoated glass slides, and then deplasticized. The sections were evaluated for cellular evidence of DNA fragmentation indicative of apoptosis, employing the TUNEL methodology. Following a 15-min digestion at room temperature with 1 mg/mL proteinase K, the endogenous peroxidase activity was quenched with 3% H_2_O_2_ solution. Terminal deoxynucleotidyl transferase and digoxigenin-labeled dUTP were introduced after the sections had been placed in an equilibration buffer, and incubation occurred for 1 h at 37 °C. After being rinsed in stop-wash buffer, the TUNEL signal was identified with a peroxidase-labeled antidigoxigenin antibody, followed by development with diaminobenzidine. Eight randomly selected fields were analyzed for each section. The resulting images were captured, and the proportion of apoptotic cells within each section was computed with Image-Pro Plus 1.42 software (Media Cybernetics, Silver Spring, MD, USA) by calculating the ratio of TUNEL-positive nuclei to the total number of nuclei.

### 2.11. Immunohistochemistry

Sections were sliced to a thickness of 4 um and placed on glass slides coated with protein. The assessment of cleaved caspase-3 expression in paraffin-embedded sections was conducted through immunohistochemical staining of mouse osteolytic interface periosteum tissues. Following the quenching of endogenous peroxidase activity, antigen retrieval, and blocking of nonspecific binding sites, the sections were incubated overnight at 4 °C with primary antibodies targeting cleaved caspase-3. After performing three washes, ChemMateTM EnVisionTM/HRP (Agilent Technologies, Santa Clara, CA, USA) was applied, and the sections were incubated for 30 min at room temperature. Staining was subsequently developed using a 3,3′-diaminobenzidine substrate (DABS) solution, followed by counterstaining with hematoxylin. The stained sections were evaluated to determine the percentage of positively stained cells for the cleaved caspase-3 in osteolytic interface periosteum tissues. Images were captured, and the total count of marker-positive cells was calculated through Image-Pro Plus software.

### 2.12. Statistical Analysis

Data are presented as the mean ± standard error of the mean (SEM). Differences in the mean values of variables of the experimental parameters among the groups were evaluated using analysis of variance (ANOVA). Post hoc testing of differences between groups was performed by using Duncan’s test when ANOVA was significant. A *p*-value < 0.05 was considered indicative of a significant difference. No adjustment for multiple testing was applied since the statistical analysis was performed in an exploratory way. Statistical analyses were processed with the SPSS v17.0 software package (SPSS Inc, Chicago, IL, USA).

## 3. Results

### 3.1. 4-PBA Ameliorates the Histopathologic Change of Osteolysis

In the murine model employed for the study of osteolysis, no mortality was observed either during or after the implantation of TiNPs. The mice retained normal activity levels throughout the experiment. Two weeks after calvaria bone implantation of TiNPs, the mice were euthanized, and the histopathologic changes related to PIO were evaluated by TB, HE, and TRAP staining.

The implantation of TiNPs led to a notable increase in the number of TB-positive bone resorption lacunas observed in the calvarial caps (Figure 1A,B), suggesting the occurrence of significant osteolysis within the affected areas. The resorption ratios of the lacunas in the presence of TiNPs were 12.6 times higher when compared to those in the Control group. Conversely, the osteolytic areas saw a remarkable reduction with the combined intervention of 4-PBA. Specifically, the co-intervention of 4-PBA resulted in a 58% decrease in the bone resorption ratio among the TiNP-treated mice, highlighting the therapeutic potential of 4-PBA in mitigating the adverse effects of TiNPs on bone integrity.

The HE-stained osteolytic calvarial caps further corroborated the finding that the co-intervention of 4-PBA effectively mitigated TiNP-induced bone erosion (Figure 1C,D). The implantation of TiNPs resulted in a significant increase in inflammatory infiltrates, as evidenced by the extensive infiltration of inflammatory cells along the eroded surfaces of the bone when compared to the Control group. Furthermore, the morphology of the bone was preserved in the Control group, whereas a rise in the surface area of bone erosion was noted in osteolytic bone tissues following implantation of TiNPs. This suggests that the introduction of TiNPs had a detrimental impact on the bone’s structural condition, leading to increased erosion in the affected calvarial caps. Conversely, the chaotic bone morphology, extensive bone erosion, and presence of infiltrating inflammatory cells were markedly reduced with the co-intervention of 4-PBA.

To determine if the in vivo effects of osteolysis correlated with an increase in osteoclast numbers, we subsequently assessed and contrasted the quantity of osteoclasts. The outcomes of TRAP histochemical staining (Figure 1E,F), which serves as a cytochemical indicator for osteoclasts, aligned with the findings from both the TB and HE staining methods. Osteoclasts appeared on the calvarial caps following the implantation of TiNPs. Notably, the presence of 4-PBA as a co-intervention led to a 59% decrease in the osteoclast count within the TiNP-treated mice.

### 3.2. 4-PBA Promotes Particle-Induced Macrophage Apoptosis in Osteolytic Interface Periosteum

To verify the presence of macrophage apoptosis in osteolytic interface periosteum, we employed the TUNEL method to identify cells showing signs of apoptosis (Figure 2A,B). Compared to the Controls, there was a notably greater percentage of apoptotic macrophages following TiNP implantation. Additionally, the experimental findings unveiled a discernible positive correlation between the severity of osteolysis and the incidence of apoptosis. Nonetheless, the cells with TUNEL labeling were most pronounced in areas where osteolysis remission occurred due to the combined treatment of TiNPs and 4-PBA. Moreover, the intervention with 4-PBA alone resulted in only a limited number of positive cells when compared to the Control group and tissues stimulated by particles.

We further examined the presence of cells exhibiting positive reactions for cleaved caspase-3 via immunohistochemistry (Figure 2C,D), as well as total protein expression levels (Figure 3A,B) and enzyme activity (Figure 4A) of caspase-3 in osteolytic interface periosteum. The expression levels and enzyme activity of caspase-3, along with the average proportion of macrophages exhibiting cleaved caspase-3, increased with the implantation of TiNPs. Nonetheless, the significant effect was even more pronounced in the area where TiNPs were combined with the 4-PBA intervention.

In the group receiving TiNPs, histomorphological analysis revealed the presence of not only apoptotic macrophages but also an appropriate quantity of fibroblasts. Conversely, the group treated with TiNPs in conjunction with 4-PBA exhibited a significant decrease in both the number of normal macrophages and fibroblasts, accompanied with a marked increase in apoptotic macrophages. Consequently, compared to the TiNPs group, the proportion of apoptotic macrophages within the total cell population in the TiNPs + 4-PBA group increased markedly accordingly. This increase in apoptosis was accompanied by a decline in the proliferative capacity of both fibroblasts and macrophages, suggesting a potential adverse effect of the combined treatment on the viability, survival, and pro-inflammatory activities of these stress-damaged cells.

### 3.3. Regulators of Apoptosis (Bcl-2 and Bax) in 4-PBA-Induced Macrophage Apoptosis Within Osteolytic Interface Periosteum

To investigate the potential involvement of Bcl-2 and Bax in 4-PBA-induced macrophage apoptosis within osteolytic interface periosteum, we assessed the expression levels of Bax (Figure 3A,C) and Bcl-2 (Figure 3A,D). The pro-apoptotic protein Bax demonstrated significantly higher activation levels in the osteolytic interface periosteum exposed to TiNPs when compared to the Control group. Conversely, the expression of the anti-apoptotic protein Bcl-2 was notably downregulated in TiNP-exposed group relative to the Control group. Nonetheless, pharmacological intervention with 4-PBA facilitated apoptosis induced by TiNPs by elevating the levels of Bax and reducing Bcl-2 levels when compared to the TiNPs group.

### 3.4. Reduction of Inflammatory Factors Facilitated by 4-PBA in Osteolytic Interface Periosteum

To explore the inflammatory factors underlying the impact of 4-PBA co-intervention on TiNP-induced osteolysis, we monitored the expression levels of several pro-inflammatory cytokines (TNF-α, IL-1β, and IL-6) (Figure 4B–D) and the production of ROS (Figure 4E). The presence of TiNPs resulted in considerable inflammation, as evidenced by a notable rise in TNF-α, IL-1β, IL-6, and ROS levels when compared to the negative Control. Conversely, the simultaneous intervention of 4-PBA markedly decreased the expression of inflammatory cytokines and the production of ROS.

### 3.5. Reduction of Osteoclastogenic Cytokines Facilitated by 4-PBA in Osteolytic Interface Periosteum

A significant increase in TRAP activity was observed following the exposure of the periosteum to TiNPs, while the co-intervention of 4-PBA led to a reduction in TRAP activity (Figure 4F). As illustrated in Figure 4G,H, the implantation of TiNPs resulted in a substantial increase in M-CSF and RANKL levels, with increases of 24.3-fold and 5.2-fold, respectively, compared to the Control group. In contrast, the concurrent intervention of 4-PBA resulted in a marked reduction decrease in these osteoclastogenic cytokines. Specifically, co-intervention with 4-PBA led to a 77% reduction in M-CSF levels and a 61% decrease in RANKL levels compared to the TiNPs group.

## 4. Discussion

The interface membranes surrounding loosening prostheses are highly enriched with macrophages, constituting 60–80% of the total cell composition [13]. Crucially, previous clinical and in vitro studies have consistently observed the presence of apoptotic macrophages in these osteolytic interface membranes, with their quantity escalating with disease progression [13,14,16]. This programmed cell death is posited as a protective mechanism against particle-induced stress, emphasizing its critical role in PIO [13,16]. Therefore, understanding the interplay between macrophage apoptosis and the progression of PIO is vital for comprehending the processes of bone resorption and destruction [45,46]. Accordingly, our study specifically investigated the effects of TiNPs (average 51.7 nm, consistent with particle sizes found in retrieved tissues [14,47]) on a murine calvaria resorption model, focusing on inflammatory responses and macrophage behavior, including osteoclastogenesis and apoptosis.

Currently, no specific drug is approved for the prevention or inhibition of periprosthetic osteolysis [48,49,50]. However, modulating osteoclast formation/function and reducing inflammation are promising therapeutic targets [0, 96]. Considering that the increased recruitment of macrophages due to inflammation, along with their subsequent transformation into osteoclasts, significantly contributes to the advancement of PIO [11,46,51,52,53], we propose that a diminished number of osteoclasts may be triggered either by a reduction in cytokine secretion or by the induction of apoptosis in osteoclast precursor cells, which are, specifically, macrophages. Therefore, therapeutically inducing macrophage apoptosis through the administration of specific pharmacological drugs, while avoiding the instigation of an inflammatory response, could serve as a promising strategy for mitigating macrophage responses to wear particles [13,14,16].

Recent research has indicated that 4-PBA functions as a chemical chaperone within the ER and plays a protective role in the phenotypic features of osteoclasts, as well as exhibiting anti-inflammatory properties [29]. The evident occurrence of ERS in PIO, along with the suppression of osteoclast activity and the anti-inflammatory properties of 4-PBA [7,8,17,26,27,39,40], makes it reasonable to postulate that the use of the chemical chaperone 4-PBA to inhibit ERS may have therapeutic benefits in ameliorating bone resorption and the overall progression of osteolysis. While 4-PBA’s mechanisms, including histone deacetylase inhibition and caspase activation, are understood in various cell types [42], its precise effects on macrophage apoptosis in osteolytic environments induced by wear particles remained to be fully clarified. This study aimed to specifically address whether 4-PBA could alleviate PIO by promoting macrophage apoptosis, without triggering an inflammatory response.

The murine osteolysis model provides compelling evidence indicating that TiNPs can induce osteolysis, activate inflammatory responses, and encourage the differentiation of osteoclasts, accompanied by a significant increase in apoptosis at the periosteum of the osteolytic interface (Figure 5). These outcomes aligned with previous findings in mouse models of PIO and clinical samples related to prosthesis loosening [7,8,17,26,27,39,40]. These findings shed light on the biological mechanisms underpinning the effects of wear particles, highlighting their potential impact on bone health and the inflammatory processes linked to bone resorption and implant failure. Consistent with previous findings [26,27,39], mice experiencing PIO that were intervened with the 4-PBA exhibited a significant reduction in the severity of osteolysis, a diminished extent of bone erosion and damage, a lesser degree of disorganized bone structure, decreased levels of inflammatory cytokines and ROS, a lower frequency of inflammatory cell infiltration, a reduced count of osteoclasts, and decreased production of osteoclastogenic cytokines. Nevertheless, the extent of apoptosis, in conjunction with the apoptosis-regulating proteins Bcl-2 and Bax, were notably accentuated after the co-intervention with 4-PBA. This observation suggests that the osteolytic interface periosteum environment, induced by TiNPs, exhibited an increased tendency for apoptosis as a result of the pharmacological intervention provided by 4-PBA. Such an alteration in the apoptotic activity indicates that the inhibition of ERS is involved in the modulation of the apoptotic responses triggered by particles and contributes positively to the alleviation of bone defects and the restoration of bone integrity in mice experiencing PIO.

The osteolytic cascade, initiated by cytokine release from macrophages, has garnered a significant amount of research attention [11]. However, the related phenomenon of apoptosis induction within this particular context has been explored by only a limited number of researchers. Thus far, in vitro studies have demonstrated that wear particles (ceramic, metal, and polyethylene particles) possess the capability to induce apoptosis in macrophages [14,54]. Additionally, instances of macrophage apoptosis have not only been observed in laboratory settings but also in the actual clinical environment [13,15,16,18]. Specifically, such apoptosis has been noted in periprosthetic locations, which encompass the capsules and interface membranes of patients experiencing aseptic loosening of hip implants [13,16]. Previous research has demonstrated a significant correlation between the severity of osteolysis and an increased incidence of macrophage apoptosis [13]. This connection suggests that as the level of osteolysis intensifies, the rate of cellular apoptosis also rises. Furthermore, the mechanisms through which macrophage apoptosis occurs in this disease process involve a comprehensive activation of all three well-established apoptosis pathways, namely the death receptor pathway, the ERS pathway, and the mitochondrial pathway [14,16]. Previous findings indicated that inducing apoptosis in macrophages through the use of bisphosphonates or other pro-apoptotic agents could serve as an effective therapeutic strategy to diminish the macrophage response to wear particles [20,21]. Focusing on macrophage apoptosis may provide a logical and scientifically grounded strategy for both preventing and treating various conditions, like inflammatory osteolysis and aseptic loosening.

In this study, we identified that the pharmacological intervention with 4-PBA significantly ameliorates PIO by facilitating macrophage apoptosis in interface periosteum tissues affected by osteolysis. The identification of macrophage apoptosis through pharmacological intervention with 4-PBA during the progression of PIO may provide a desirable therapeutic endpoint for treatment. Nevertheless, what kind of role do the apoptotic macrophages play in the pathogenesis of osteolysis and the resulting aseptic loosening? To investigate the underlying therapeutic impact of encouraging macrophage apoptosis in alleviating osteolysis more deeply, a comprehensive analysis was conducted, incorporating valuable insights gleaned from prior studies within this field [13,14,16,30,37,41,42,54,55,56,57]. By synthesizing this body of knowledge, the study aims to elucidate the mechanisms through which promoting macrophage apoptosis may contribute to the mitigation of osteolytic conditions.

4-PBA is recognized for its significant function as a chemical chaperone that assists in reducing stress in the ER [58]. Activation of ERS has been documented to play a significant role in PIO, while the inhibition of ERS with 4-PBA ameliorated ERS ameliorated the severity of osteolysis across these different inflammatory scenarios [7,8,17,26,27,39,40]. The response to ERS can facilitate apoptosis and enhance inflammation [59,60,61]. Interestingly, the use of 4-PBA to inhibit this stress response decreases inflammation while simultaneously increasing apoptosis. The suppression of ERS signaling by 4-PBA has the effect of reducing inflammation, which could be beneficial in various pathological contexts where chronic inflammation is a concern [26]. However, this reduction in inflammation comes at a cost, as it also leads to an increase in apoptosis. This dual effect emphasizes the intricate balance the ERS response maintains between survival and cell death, suggesting that while targeting this pathway may alleviate inflammation, it could inadvertently heighten the risk of apoptosis, thus impacting overall cell viability.

The occurrence of increased apoptosis resulting from the alleviation of ERS might appear paradoxical, considering that the ERS is commonly linked to the activation of apoptotic pathways [59,60,61]. However, recent research has substantiated this phenomenon that the relief of ERS can trigger cell cycle arrest and apoptosis via other apoptotic mechanisms apart from ERS itself [30,41], including the JNK pathway [42], mitochondrion pathway [43], or death receptor activation [44]. Such findings highlight the complexity of cellular responses to ERS and suggest that relieving this stress does not necessarily promote cell survival; instead, it can activate alternative pathways that culminate in programmed cell death. Nevertheless, the idea that reducing ERS might simultaneously elevate the rate of apoptosis challenges conventional understanding, underscoring the complexity of apoptotic regulation amidst ERS [60]. This dynamic interaction between cellular stress responses and the mechanisms that govern cell survival and death warrants further investigation to elucidate the underlying mechanisms involved.

The insights into the processes surrounding macrophage apoptosis and the remission of inflammation have been extensively documented in the existing literature [13]. Numerous studies highlight that in the context of PIO, there is a notable pathological increase in the apoptosis of macrophages [14,16]. This particular phenomenon is not isolated, as it has also been observed in various other diseases, suggesting a potential commonality in the underlying mechanisms at play [62,63,64]. In such instances, the extent and intensity of apoptotic reactions are so great that the remnants of the affected cells cannot be entirely eliminated from the tissue [13]. Consequently, this incomplete removal of cellular debris subsequently contributes to an elevated release of fibrogenic mediators, which could be responsible for the increased proliferation of fibroblasts in spite of the increased apoptosis [13]. Moreover, this process results in a heightened excretion of inflammatory factors that may trigger the activation of osteoclasts and the recruitment of additional macrophages, potentially worsening the advancement of the inflammatory condition [13]. As previously mentioned [14,25,26], the progression and escalation of periprosthetic osteolysis are often associated with an increase in macrophage apoptosis, along with elevated levels of inflammatory factors in the surrounding environment. Notably, the most important discovery in our study is that the inhibition of ERS through 4-PBA considerably ameliorates the severity of osteolysis, lessens the inflammatory response, and concurrently increases the rate of macrophage apoptosis. It may appear counterintuitive that the alleviation of ERS can result in a heightened rate of apoptosis, even as there is a concomitant decline in inflammation. This observation challenges traditional notions about the relationship between stress responses within cells and the processes of cell death and inflammation [65]. The results, along with the observed dual role of 4-PBA [30,41,42,55], underscore the complex interplay among ERS regulation, the inflammatory response, and the survival of macrophages in osteolytic disease conditions. This complexity implies that interventions targeting ERS and inflammation may have diverse and intricate effects on cellular outcomes and the overall disease progression, highlighting the importance of a nuanced understanding of these processes in crafting effective therapeutic strategies for conditions where ERS and inflammation are both contributors to disease progression.

In light of the findings delineated in this study, along with the insights acquired from the existing literature [30,55], we clarify the underlying mechanism through which 4-PBA facilitates apoptosis and reduces the inflammatory response in macrophages. Apoptosis is a highly regulated and active process of cell death that serves a critical function in the physiological turnover of normal cells, thereby contributing to the maintenance of tissue homeostasis [60,66]. This process allows the body to precisely control the number of cells within various tissues, ensuring that they remain healthy and functional [61]. Moreover, apoptosis is vital for removing cells that could pose a threat to the organism’s overall survival, such as damaged or potentially cancerous cells [66]. By facilitating the elimination of these undesirable cells, apoptosis not only preserves tissue integrity but also plays a significant role in protecting the organism from disease and promoting overall health [67]. Pharmacological intervention with 4-PBA presents a promising opportunity as a therapeutic agent for the management of macrophage health, particularly in cases where these immune cells are impacted by particulate matter [26]. By addressing the effects of particles on these macrophages, 4-PBA might aid in enhancing immune responses and promoting better health outcomes for individuals affected by such exposures [27,39]. The compound has the potential to facilitate programmed cell death, which, in conjunction with its ability to regulate the activation of these compromised and distressed cells, could assist in re-establishing a state of immune equilibrium [58]. Through these mechanisms, 4-PBA may help to diminish inflammation, reduce tissue degradation, and alleviate the detrimental effects resulting from particle-induced osteolytic damage in the surrounding tissues [29]. The 4-PBA approach seeks to leverage the biological mechanisms of these immune cells to either inhibit unwanted apoptotic processes or encourage beneficial ones [38,58]. This dual strategy is intended to optimize the functioning of immune cells, ultimately leading to improved health outcomes [41,55].

It has been established that the promotion of apoptosis in stress-damaged macrophages, which manage to survive without undergoing apoptosis and exhibit characteristics distinct from normal macrophages, could serve as a valuable therapeutic target for alleviating aseptic osteolysis and thwarting prosthetic loosening [13,16]. Fortunately, 4-PBA possesses the essential features and capabilities that allow it to effectively address the challenges presented in our study of inflammatory osteolysis, as well as in other inflammatory conditions explored in prior studies [30,41,42], reinforcing its significance in the field of medical research. The process involving 4-PBA enables the body to accurately regulate the quantity of cells present, effectively identifying and eliminating those that have sustained stress-induced damage. Such stress-damaged cells can potentially jeopardize the overall health and survival of the organism. By facilitating this selective removal, 4-PBA contributes to the maintenance of a population of healthy and functional cells, which is essential for the organism’s well-being and resilience against adverse conditions [30,41]. Moreover, 4-PBA exhibits relatively low toxicity levels, demonstrates stability, is generally well-accepted, and has been clinically evaluated for safety [41]. Its use can be sustained on a daily basis and even throughout a lifetime, as it is naturally synthesized within the intestinal tracts of humans and animals alike [41]. Above all, it is noteworthy that the intervention with 4-PBA, when administered as a standalone treatment in the absence of wear particles, exhibited a relatively low level of toxicity and was deemed safe based on experimental observations. In addition, this intervention did not appear to elicit any notable alterations in the inflammatory mediators of macrophages or the inflammatory response. This discovery is particularly important, as it suggests that 4-PBA can potentially be utilized in various therapeutic contexts without posing a substantial risk to health, at least during the initial stages of evaluation.

While our in vivo study protocols align with extensively used models in PIO research [18,26,27,68,69,70], several limitations warrant consideration. Firstly, our study focused solely on nanoscale TiAl6V4 metal particles. Although TiNP-induced osteolysis serves as a prevalent model for evaluating pharmaceutical effects in PIO and its mechanisms are comparable to those of other wear particles [71], future comprehensive validation should explore biological responses to different particle sources and types. Secondly, the murine calvaria resorption model, though widely used to replicate periprosthetic osteolysis [18,26,27,68,69,70,72], represents an acute pathological condition (2-weeks duration) rather than a chronic clinical scenario. Furthermore, it does not incorporate a genuine prosthetic device. Therefore, supplementary investigations employing alternative strategies are needed for thorough validation. Thirdly, while 4-PBA effectively alleviates particle-induced ERS [26,27], the extent of its effective delivery to the calvaria and the specific region of TiNPs application remains to be definitively defined. Furthermore, the persistence of 4-PBA’s effects after treatment requires further investigation to fully understand its therapeutic implications. Fourthly, the lack of a positive-control group represents a limitation of our study. However, this reflects a broader challenge in the field, as there are currently no FDA-approved pharmacological therapies specifically for the prevention and management of periprosthetic osteolysis, a challenge noted in most drug intervention studies for PIO. Fifth, while we have demonstrated that 4-PBA reduces PIO by promoting overall tissue apoptosis, the specific downstream mechanisms by which it exclusively promotes macrophages apoptosis and the exact signaling pathways involved in this process remain to be fully elucidated. Further investigation, potentially incorporating macrophage-specific markers or cell sorting techniques, is necessary to clarify these mechanisms and enhance our understanding of how 4-PBA exerts its protective effects in this context. Ultimately, it is essential to recognize that wear particles can interact with various cell types, resulting in complex and frequently detrimental reactions at both cellular and molecular levels. Consequently, future research should focus on investigating the effects of 4-PBA on macrophages, along with other significant cell types, such as osteoblasts, osteoclasts, osteocytes, and fibroblasts, in in vitro settings. Conducting such studies will contribute to a more thorough understanding of how 4-PBA may influence biological reactions within the periprosthetic microenvironment, potentially revealing its role in modulating local cellular responses and improving the overall outcomes for patients with implants.

## 5. Conclusions

The presence of wear particles can trigger a series of biological processes, including an inflammatory reaction, osteoclastogenic response, and macrophages apoptosis, as observed in a murine model of TiNP-induced osteolysis. This sequence of events significantly contributes to the progression of inflammatory osteolysis. In this context, pharmacological intervention with 4-PBA has demonstrated potential in ameliorating the severity of osteolysis induced by these particles. This therapeutic agent functions by inhibiting the inflammatory response, thereby reducing the activation and differentiation of osteoclasts, while simultaneously promoting the apoptosis of macrophages. The chemical chaperone 4-PBA, which exhibits low toxicity and selectivity toward stress-damaged cells, presents a promising new approach for addressing inflammatory osteolysis that is refractory to conventional anti-bone resorption agents. This approach, which aims to enhance the longevity and efficacy of orthopedic prosthetic devices, could potentially be instrumental in preventing osteolysis and aseptic loosening—major complications that can occur following the implantation of joint prostheses. While the pathogenic role of increased macrophage apoptosis, in conjunction with heightened proliferation and the intricate interactions among ERS regulation, the inflammatory response, and macrophage survival in periprosthetic osteolysis requires further elucidation, it suggests a promising avenue for ongoing research and innovation, as the potential benefits of 4-PBA could be harnessed effectively while minimizing adverse health effects by specifically targeting the affected cells.

## Figures and Tables

**Figure 1 bioengineering-12-00701-f001:**
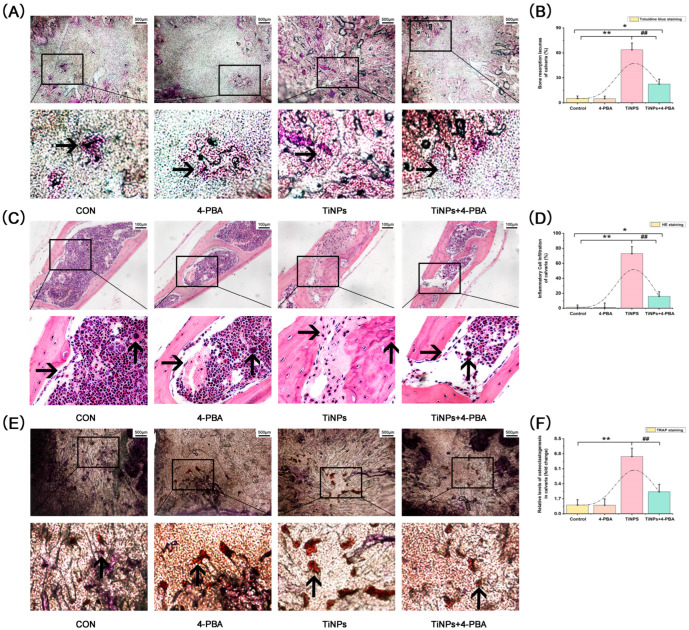
The histopathologic change in osteolysis in the murine calvaria resorption model. (**A**,**B**) Histological analysis of calvaria stained with toluidine blue (TB). Bone resorption lacunas (
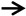
) were identified according to histomorphological criteria. Original magnification: ×40. Scale bar: 500 μm. (**C**,**D**) Histological analysis of calvaria stained with hematoxylin and eosin (HE). The infiltration of inflammatory cells (
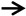
) and osteoclasts (
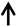
) in calvaria was observed. Original magnification: ×200. Scale bar 100 μm. (**E**,**F**) Histological analysis of calvaria stained with tartrate-resistant acid phosphatase (TRAP). Osteoclasts (
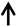
) were identified according to histomorphological criteria. Original magnification: ×40. Scale bar 500 μm. The quantitative data (**B**,**D**,**F**) are derived from unbiased histomorphometric analysis of multiple fields and samples (*n* = 5 mice per group). The images are representative fields, and the quantification reflects the average of multiple fields and samples. Data are represented as the means ± S.E.M from three independent experiments. *: *p* < 0.05, **: *p* < 0.01 versus Control. ^#^: *p* < 0.05, ^##^: *p* < 0.01 versus TiNPs.

**Figure 2 bioengineering-12-00701-f002:**
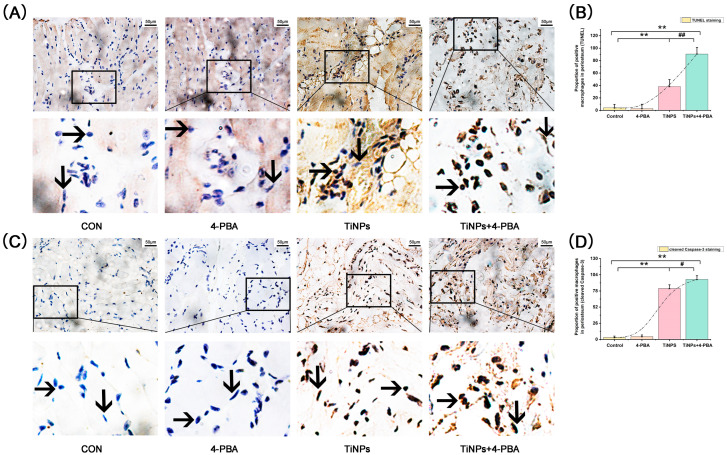
4-PBA promotes particle-induced macrophage apoptosis. (**A**,**B**) Macrophages exhibiting positive TUNEL staining in osteolytic interface periosteum. (**C**,**D**) Cleaved caspase-3 positive macrophage in osteolytic interface periosteum. Macrophages (
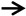
) and fibroblasts (
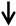
) were identified according to histomorphological criteria. Original magnification: 400×. Scale bar: 50 μm. The quantitative data (**B**,**D**) are derived from unbiased histomorphometric analysis of multiple fields and samples (*n* = 5 mice per group). The images are representative fields, and the quantification reflects the average of multiple fields and samples. Data are represented as the means ± S.E.M from three independent experiments. *: *p* < 0.05, **: *p* < 0.01 versus Control. ^#^: *p* < 0.05, ^##^: *p* < 0.01 versus TiNPs.

**Figure 3 bioengineering-12-00701-f003:**
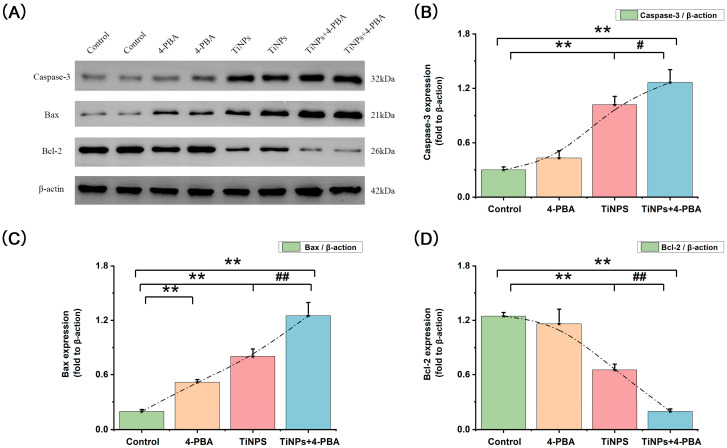
The expression levels of apoptosis-related protein in osteolytic interface periosteum. (**A**–**D**) Western blots of caspase-3, Bax, and Bcl-2 in osteolytic interface periosteum. The density of the Western blot bands was quantified using ImageJ software. Data are represented as the means ± S.E.M from three independent experiments. *: *p* < 0.05, **: *p* < 0.01 versus Control. ^#^: *p* < 0.05, ^##^: *p* < 0.01 versus TiNPs.

**Figure 4 bioengineering-12-00701-f004:**
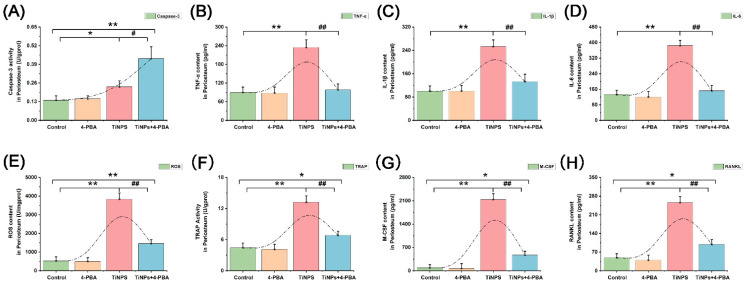
Caspase-3 enzyme activity, inflammatory factors, and osteoclastogenic cytokines in vivo osteolytic interface periosteum tissues. (**A**) The levels of caspase-3 enzyme activity. (**B**–**D**) The expression levels of pro-inflammatory cytokines (TNF-α, IL-1β, and IL-6). (**E**) The production of ROS. (**F**–**H**) The levels of TRAP, M-CSF, and RANKL. Data represent as the means ± SEM from three independent experiments. *: *p* < 0.05, **: *p* < 0.01 versus Control. ^#^: *p* < 0.05, ^##^: *p* < 0.01 versus TiNPs.

**Figure 5 bioengineering-12-00701-f005:**
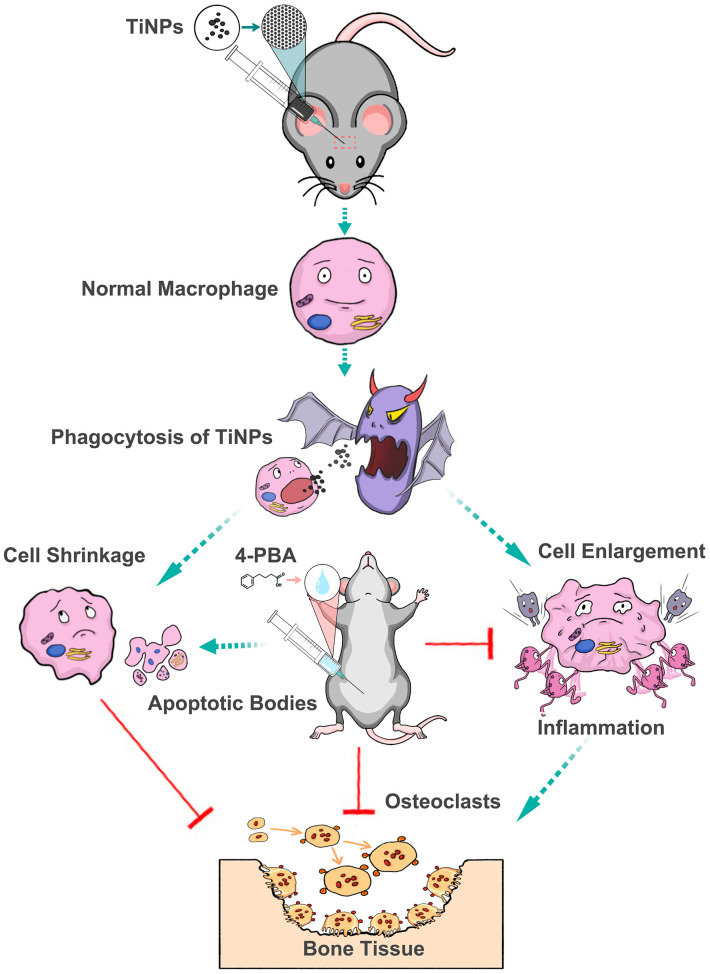
Pharmacological intervention with 4-phenylbutyrate ameliorates TiNP-induced inflammatory osteolysis by suppressing the inflammatory response, decreasing the activation and differentiation of osteoclasts, and concurrently promoting the apoptosis of stress-damaged macrophages in a murine calvarial resorption model.

## Data Availability

The raw data supporting the conclusion of this article will be made available from the first author.

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
