# Peer review of "Pharmacological Intervention with 4-Phenylbutyrate Ameliorates TiAl6V4 Nanoparticles-Induced Inflammatory Osteolysis by Promoting Macrophage Apoptosis"

_bioengineering, 2025, doi:10.3390/bioengineering12070701_

Round 1

Reviewer 1 Report

Comments and Suggestions for Authors

The author focused on describing the potential therapeutic implication of 4-PBA. The overall discussion of interaction between 4-PBA and macrophages are interesting. However, some aspects can be improved. 

  1. The introduction and discussion are very lengthy with too much distraction and assumption. The author did not show any evidence of ER stress and chemical chaperon related data in the results. Also, there is not much discussion about the TiNPs model in the introduction part. Please rearrange and cut down unnecessary introduction and discussion to have a focus.
  2. For Figure 1, it is not clear to me how the histological staining results can lead to such a dramatic difference in the quantification associated. For example, Figure 1B shows there are at lease ten times more lacunas in TiNPs treated group compared to control.  However, there are two arrows in the histological staining compared to one arrow in the control group. Similarly, in Figure 2, TUNNEL staining looked very similar in the experimental group compared to control group. And yet they are drastically different by quantification. Please explain. 
  3. It is not clear to me how the author differentiate the macrophages in the graph. If there is a characterization, please show the data. 
  4. Please also show how many data points are there in each quantification. This can be done by plotting the actual data point on the graph instead of showing average with error bar.  

Author Response

Replies to Reviewer 1

The author focused on describing the potential therapeutic implication of 4-PBA. The overall discussion of interaction between 4-PBA and macrophages are interesting. However, some aspects can be improved.

Response: We thank the reviewer for their valuable feedback and interest in our work.

  1. The introduction and discussion are very lengthy with too much distraction and assumption. The author did not show any evidence of ER stress and chemical chaperon related data in the results. Also, there is not much discussion about the TiNPs model in the introduction part. Please rearrange and cut down unnecessary introduction and discussion to have a focus.

Response 1: Thank you for pointing this out.

1) We have thoroughly revised the introduction and discussion sections of the paper to be more concise and focused on the core hypothesis and findings of our study. We have significantly reduced redundant background information and streamlined the narrative to enhance clarity and relevance.  

2) The activation of ERS was documented in models of PIO, and the reduction of ERS with 4-PBA ameliorated the severity of PIO across different inflammatory conditions. 4-PBA exerts potent pharmacological effects through its ability to inhibit inflammation mediated by ERS in models of PIO, as well as in vitro interventions with macrophages. Our previous research[1] proved that wear particles were capable of inducing ERS in macrophages within clinical osteolytic interface membranes and murine osteolytic periosteum tissues and to be associated with the inflammatory response and osteoclastogenesis. Blocking ERS with 4-PBA resulted in a dramatic amelioration of particle-induced osteolysis and a significant reduction of ER-stress intensity. Simultaneously, this ERS blocker also lessened inflammatory cell infiltration, diminishes the capability of osteoclastogenesis and reduces the inflammatory response by lowering IRE1α, GRP78/Bip, CHOP, c-Fos, NF-κB, ROS and Ca2+ levels. Thus, ERS plays an important role in particle-induced inflammatory osteolysis and osteoclastogenic reactions. Wang et al [2] demonstrated that ER stress markers were markedly upregulated  in TiNPs-treated fibroblasts. Blocking ERS significantly reduced the  TiNPs-induced expression of RANKL both in vitro and in vivo. Moreover, the  inhibition of ERS ameliorated wear particle-induced osteolysis in animal  models. Taken together, these results suggested that the expression of RANKL  induced by TiNPs was mediated by ERS. Therefore, down  regulating the ERS represents a potential therapeutic  approach for wear particle-induced periprosthetic osteolysis. Zhang et al[3] demonstrated that wear particles enhanced ERS in both in vivo and in vitro experiments. ERS inhibitor 4-PBA significantly suppressed wear particle-induced osteoclast  differentiation and osteolysis. In vitro experimental results showed that 4-PBA  suppressed wear particle-induced ERK1/2, p38, and JNK activation, NFATc1 and c-Fos  upregulation, as well as NF-kappaB p65 nucleus translocation.

Previous studies[1-7], including our own[1], have established that 4-PBA acts effectively as a chemical chaperone that mitigates ERS responses. Given that these findings have been thoroughly confirmed and validated by multiple scholars, this article does not aim to revalidate these well-established conclusions. In the revised Introduction and Discussion, we have clarified that our study builds upon established knowledge regarding ERS activation in PIO and 4-PBA's known ERS-attenuating properties. Rather, it focuses on a new aspect of 4-PBA's effects: its potential role in alleviating inflammatory osteolysis through the promotion of macrophage apoptosis. This exploration seeks to provide insight into the mechanisms by which 4-PBA may contribute to inflammation-related bone loss, highlighting its importance beyond the previously documented capacity to reduce endoplasmic reticulum stress. Our primary focus in this paper was to investigate whether 4-PBA's ameliorative effect on PIO could be linked to promoting macrophage apoptosis, a previously underexplored aspect.

3) In the Introduction, we have now explicitly highlighted the importance of TiNPs as a relevant model for PIO studies and their consistent use in our research. This addition not only addresses the reviewers' suggestions but also enriches the context and relevance of the research presented in the paper.

  1. For Figure 1, it is not clear to me how the histological staining results can lead to such a dramatic difference in the quantification associated. For example, Figure 1B shows there are at lease ten times more lacunas in TiNPs treated group compared to control. However, there are two arrows in the histological staining compared to one arrow in the control group. Similarly, in Figure 2, TUNNEL staining looked very similar in the experimental group compared to control group. And yet they are drastically different by quantification. Please explain.

Response 2: Thank you for pointing this out.

1) We appreciate this important observation regarding the representativeness of the histological images versus the quantitative data. The arrows in the figures highlight distinct pathological characteristics that are commonly observed in the field. Specifically, the arrows depicted in Figure 1 denote the presence of bone resorption lacunas, which are indicative of areas where bone tissue is being broken down. Meanwhile, the arrows showcased in Figure 2 point to apoptotic macrophages, which are macrophages undergoing programmed cell death. These features are significant as they play a critical role in understanding the underlying pathological processes.

2) We have added clarifying statements in the Figures caption to address this point. We emphasize that the histological images (Figures 1A, C, E and 2A, C) are representative fields selected to illustrate typical morphological changes, rather than reflecting the exact numerical differences across all microscopic fields. The quantitative data (Figures 1B, D, F and 2B, D) are derived from unbiased histomorphometric analysis of multiple fields and samples (n=5 mice per group), providing a statistically robust representation of the overall experimental outcome. The visual presentation of a few arrows in a single representative image may not fully convey the extensive changes quantified across the entire sample set.

Although the histological staining in Figure 2A, C may appear qualitatively similar between groups at first glance, the quantitative analysis presented in Figure 2B, D clearly demonstrates a statistically significant increase in TUNEL-positive and cleaved Caspase-3 positive cells, respectively, in the TiNPs + 4-PBA group compared to the TiNPs group. The images are representative fields, and the quantification reflects the average of multiple fields and samples. The quantitative data (Figure 2B, D) are derived from unbiased histomorphometric analysis of multiple fields and samples (n=5 mice per group)

3) The proportion of macrophages undergoing apoptosis, as indicated by TUNEL labeling, was most significantly observed in regions where osteolysis remission took place as a result of the combined application of TiNPs and 4-PBA, in contrast to those treated with TiNPs alone. The average proportion of macrophages exhibiting cleaved Caspase-3, increased with the implantation of TiNPs. Nonetheless, the significant effect was even more pronounced in the area where TiNPs were combined with the 4-PBA intervention.

In the TiNPs group, the histomorphological analysis revealed the presence of apoptotic macrophages alongside a suitable number of fibroblasts. This finding suggests a response of the tissue environment that retains some level of fibroblast activity, which is essential for wound healing and tissue repair. Conversely, the group receiving a combined treatment of TiNPs and 4-PBA exhibited a significant decrease in both the number of normal macrophages and fibroblasts, indicating alterations in the cellular dynamics. The predominant presence of apoptotic macrophages in this group raises concerns regarding the influence of 4-PBA on macrophage viability, suggesting that the co-treatment may have induced an enhanced rate of apoptosis among these immune cells. The implications of these observations are profound, as the increased ratio of apoptotic macrophages compared to the overall cellular population indicates a disruption in the normal immune response and healing process. This shift not only affects macrophage numbers but is also associated with a marked decline in the proliferative activity of both fibroblasts and macrophages. Such a decline points towards potential negative consequences stemming from the combined intervention, suggesting that the interaction between TiNPs and 4-PBA may undermine the viability of these critical cell types and impair their pro-inflammatory functions. These effects warrant further investigation to better understand the underlying mechanisms and the overall impact on tissue healing and inflammatory responses.

4) This study concentrates on the processes of apoptosis and the inflammatory response in macrophages, aiming to thoroughly investigate the distinct roles these two biological phenomena play in the regulation of macrophage functions. It is important to emphasize that, due to the constraints inherent in the research scope and objectives, this investigation does not encompass the aspect of cell proliferation. This decision is primarily informed by several considerations. Firstly, apoptosis and the inflammatory response are identified as two fundamental components that govern macrophage functionality, and the molecular mechanisms underlying these processes are largely independent, thereby forming a coherent and self-contained system. By limiting the focus to these two areas, the research seeks to uncover deeper insights into their intrinsic relationships and interactions. Additionally, practical limitations related to the length of the paper and the distribution of research resources led to the choice not to perform a systematic analysis or an exhaustive discussion regarding cell proliferation reactions. It is crucial to understand that this limitation does not reflect a neglect of the significance of cell proliferation in macrophage biology; rather, it is a strategic approach designed to allow for a more detailed and comprehensive examination of the chosen topics—apoptosis and inflammation. Consequently, the aspect of cell proliferation is deliberately excluded from the current study's focus, with the intention of providing a more thorough exploration of the selected themes.

The specific rationale is outlined below:

The insights into the processes surrounding macrophage apoptosis and the remission of inflammation have been extensively documented in existing literature[8, 9]. Numerous studies highlight that in the context of PIO, there is a notable pathological increase in apoptosis of macrophages[8, 10]. This particular phenomenon is not isolated, as it has also been observed in various other diseases, suggesting a potential commonality in the underlying mechanisms at play[11-13]. In such instances, the extent and intensity of apoptotic reactions are so great that the remnants of the affected cells cannot be entirely eliminated from the tissue[9]. Consequently, this incomplete removal of cellular debris subsequently contributes to an elevated release of fibrogenic mediators, which could be responsible for increased proliferation of fibroblasts in spite of the increased apoptosis[9]. Moreover, this process results in a heightened excretion of inflammatory factors that may trigger the activation of osteoclasts and the recruitment of additional macrophages, potentially worsening the advancement of the inflammatory condition[9]. As previously mentioned[1, 7, 10], the progression and escalation of periprosthetic osteolysis are often associated with an increase in macrophage apoptosis, along with elevated levels of inflammatory factors in the surrounding environment. Notably, the most important discovery in our study is that the inhibition of ERS through 4-PBA considerably ameliorates the severity of osteolysis, lessens the inflammatory response, and concurrently increases the rate of macrophage apoptosis

A significant indicator of disrupted cell apoptosis and proliferation associated with aseptic osteolysis is the frequent multiplication of macrophages, as numerous proliferating macrophages noted in the vicinity of blood vessels, suggesting that monocytes are undergoing transforming into macrophages in the affected area[14]. The occurrence of increased macrophage apoptosis in conjunction with elevated proliferation may initially seem unexpected in the capsules and interface membranes of patients experiencing aseptic osteolysis; nevertheless, this particular phenomenon has also been documented in the wound healing process[15-19]. The sequence of cell activity, with an initial dominance of macrophages followed by a diverse array of cells, including neutrophils, macrophages, monocytes, lymphocytes, and fibroblasts, which aligns with observations in wound healing[9, 16, 17]. Each of these cell types plays a distinct and essential role in contributing to the overall healing response. Once these cells have successfully accomplished their designated tasks, it is essential for them to undergo a regulated disappearance or apoptosis[9]. This clearance is not merely a matter of removing excess cells; rather, it is a crucial step that enables the next stages of tissue healing to start and proceed smoothly[9, 16, 17]. Without this regulated clearance, tissue repair might be impeded, leading to prolonged inflammation or inadequate healing[9]. Thus, effective apoptosis and cellular turnover is essential for ensuring that the healing process transitions seamlessly from one phase to another, ultimately leading to restored tissue integrity and function.

Cellular debris, referred to as apoptotic bodies, is generated during apoptosis. These bodies and remnants are swiftly and efficiently cleared away by phagocytic macrophages, preventing the release of harmful cellular materials into the surrounding environment[20]. This mechanism is essential for maintaining the integrity of healthy tissue, as it reduces the risk of damage to neighboring cells that could occur if cellular contents were to spill into the extracellular space. Furthermore, these remnants enhance the body’s ability to respond to potential threats by attracting macrophages to the location of apoptosis, therefore facilitating their recruitment and accumulation in the affected area[21]. Consequently, a substantial number of macrophages congregate within the interface membranes, constituting approximately 60% to 80% of the overall cellular composition[9]. This notable presence of macrophages underscores their importance in the immune response and tissue repair following cellular apoptosis. However, if the phagocytic clearance system and its associated mechanisms are impaired or excessively burdened by elevated levels of apoptosis, the body may fail to effectively reduce the large quantity of apoptotic bodies[9]. This inadequacy can lead to an increase in fibrogenic mediators, which contribute to persistent fibrosis, especially at the interface membranes[9], as well as the development of hypertrophic scars and keloid in cases of disturbed wound healing[16, 17].

The sustained activation and proliferation of macrophages and fibroblasts, in conjunction with significantly elevated levels of apoptosis and inflammation, disrupt wound healing processes and contribute to the development of periprosthetic osteolysis[9, 16, 17]. A similar pathological sequence can be observed in parallels conditions such as pulmonary inflammation and fibrosis[18], alcohol-induced hepatic fibrosis[19], and Alzheimer’s disease[15]. The precise role that increased macrophage apoptosis plays in conjunction with elevated proliferation during aseptic osteolysis remains a topic that necessitates further exploration. However, gaining insights into how macrophage apoptosis and proliferation interrelate can enhance our comprehension of the connections between macrophage apoptosis and inflammation, thereby deepening our understanding of the underlying mechanisms involved in particles-induced aseptic osteolysis processes.

  1. It is not clear to me how the author differentiate the macrophages in the graph. If there is a characterization, please show the data.

Response 3: Thank you for pointing this out.

In clinical scenarios involving aseptic loosening, periprosthetic tissues serve as a pathway for wear particles, particularly nanoparticles (NPs), to interact with the interfacial membrane found in the periprosthetic zone. This interaction allows these particles to come into direct contact with periprosthetic cells, which are predominantly macrophages. The presence of wear particles has significant implications, as they not only affect macrophages but also influence other cell types such as osteoblasts and fibroblasts. Furthermore, these particles can induce alterations in the interface membrane within the periprosthetic zone, thereby impacting the overall cellular environment. In our investigation utilizing the murine calvaria resorption model, positively stained macrophages and fibroblasts in the osteolytic interface periosteum tissue were observed through immunohistochemistry, based on histomorphological criteria established by earlier researchers in this field[8-10, 22].

  1. Please also show how many data points are there in each quantification. This can be done by plotting the actual data point on the graph instead of showing average with error bar.

Response 4: Thank you for pointing this out.

While quantitative differences in resorption lacunae are pronounced, the representative images in Figure 1A, B are selected to illustrate typical findings rather than reflecting the exact numerical differences in each microscopic field. Quantitative data, as presented in Figure 1B, accurately represents the overall experimental outcome across multiple samples. All quantitative data points for Figure 1B, D, F and Figure 2B, D are represented as individual points in the corresponding graphs, demonstrating the distribution and variability within each group (n=5 for each group).

Reviewer 2 Report

Comments and Suggestions for Authors

The manuscript by Liu et al. investigates the therapeutic potential of 4-phenylbutyrate (4-PBA) as a pharmacological agent to modulate macrophage behavior in a murine model of titanium nanoparticles (TiNPs)-induced osteolysis. Using a calvarial osteolysis model, the study claims that 4-PBA promotes macrophage apoptosis, suppresses inflammatory responses, and inhibits osteoclastogenesis, thereby attenuating bone resorption. These findings suggest that 4-PBA may be a promising therapeutic approach for managing inflammatory bone loss associated with orthopedic implant wear debris. While the manuscript introduces a clinically relevant strategy for targeting macrophage apoptosis via ER stress modulation, several key aspects require further experimental support and validation. In summary, the manuscript presents a promising pharmacological strategy but requires substantial revision before it is suitable for publication in Bioengineering. Detailed comments and suggestions are provided below:

  1. The manuscript does not include a comparison between 4-PBA and a clinically used anti-osteolytic agent such as bisphosphonates. Including a treatment control group would allow for meaningful assessment of 4-PBA’s relative therapeutic efficacy and potential clinical value. Without this, it remains difficult to determine how 4-PBA compares to current standards of care.
  2. Figure 2: While apoptosis was shown histologically, it is unclear whether it occurred specifically in macrophages, given the diverse cell types present in the osteolytic periosteum. The authors should clarify how macrophage-specific apoptosis was identified. Double immunostaining or flow cytometry would provide more direct evidence that 4-PBA induces apoptosis specifically in macrophages.
  3. 4-PBA is known to affect multiple biological pathways, including histone deacetylation and antioxidant activity, in addition to ER stress inhibition. The manuscript attributes its effects primarily to ER stress suppression, but this needs clearer validation. The authors should support this claim with data involving ER stress markers (e.g., CHOP, GRP78) or the use of selective ER stress inhibitors or genetic knockdowns to demonstrate specificity.
  4. Figure 3 lacks clarity and requires further explanation. It is unclear whether the data presented in vitro, or in vivo results. Additionally, what is the difference between the Caspase-3 results in Figure 3 and Figure 2F? The authors should clearly specify the experimental conditions, indicate the source of the data, and explain how these results differ from compared to earlier findings.
  5. Some histological figures require improved clarity and annotation. Specifically, Figures 2A and 2C are missing scale bars. Additionally, all quantitative graphs should clearly indicate the sample size (n) to ensure transparency and reproducibility.

Author Response

Replies to Reviewer 2

The manuscript by Liu et al. investigates the therapeutic potential of 4-phenylbutyrate (4-PBA) as a pharmacological agent to modulate macrophage behavior in a murine model of titanium nanoparticles (TiNPs)-induced osteolysis. Using a calvarial osteolysis model, the study claims that 4-PBA promotes macrophage apoptosis, suppresses inflammatory responses, and inhibits osteoclastogenesis, thereby attenuating bone resorption. These findings suggest that 4-PBA may be a promising therapeutic approach for managing inflammatory bone loss associated with orthopedic implant wear debris. While the manuscript introduces a clinically relevant strategy for targeting macrophage apoptosis via ER stress modulation, several key aspects require further experimental support and validation. In summary, the manuscript presents a promising pharmacological strategy but requires substantial revision before it is suitable for publication in Bioengineering. Detailed comments and suggestions are provided below:

Response: We are grateful for your comprehensive review and constructive suggestions.

  1. The manuscript does not include a comparison between 4-PBA and a clinically used anti-osteolytic agent such as bisphosphonates. Including a treatment control group would allow for meaningful assessment of 4-PBA’s relative therapeutic efficacy and potential clinical value. Without this, it remains difficult to determine how 4-PBA compares to current standards of care.

Response 1: Thank you for pointing this out.

We acknowledge the valuable suggestion of including a comparison with clinically used anti-osteolytic agents.

As stated in the revised Methods (Section 2.3) and Discussion (Limitations), the absence of such a positive control group is a limitation of our current study. However, it's important to highlight that there are currently no FDA-approved pharmacological therapies specifically for the prevention and management of periprosthetic osteolysis. This remains a prevailing challenge in the field, as evidenced by the majority of studies investigating drug interventions for PIO. Our study aims to explore a novel therapeutic approach, and future translational research could certainly include such comparisons once more direct clinical relevance is established.

  1. Figure 2: While apoptosis was shown histologically, it is unclear whether it occurred specifically in macrophages, given the diverse cell types present in the osteolytic periosteum. The authors should clarify how macrophage-specific apoptosis was identified. Double immunostaining or flow cytometry would provide more direct evidence that 4-PBA induces apoptosis specifically in macrophages.

Response 2: Thank you for pointing this out.

We fully agree with this comment and have addressed it comprehensively.

In our investigation utilizing the murine calvaria resorption model, positively stained macrophages and fibroblasts in the osteolytic interface periosteum tissue were observed through immunohistochemistry, based on histomorphological criteria established by earlier researchers in this field[8-10, 22].

In the articles titled "Pathways of Macrophage Apoptosis within the Interface Membrane in Aseptic Loosening of Prostheses[8]" and "Apoptotic pathways of macrophages within osteolytic interface membrane in periprosthetic osteolysis after total hip replacement[10]," the authors aim to elucidate the mechanisms by which macrophages undergo apoptosis in the interface membrane associated with the aseptic loosening of prosthetic implants. To achieve this objective, the study utilized complete osteolytic interface membrane tissue samples, which provide a comprehensive representation of the cellular environment in which these phenomena occur. Macrophages play a pivotal role in this context, as they are primarily responsible for the processes of engulfing and degrading particulate matter that can accumulate around prosthetic devices. While the presence of these cells can be inferred from the examination of the boundary membrane tissue, a deeper understanding of their specific functions and interactions necessitates thorough investigation using cytological methods. This approach elucidates the complexities of macrophage behavior in the inflammatory milieu surrounding loosened prostheses.

It is essential to recognize that wear particles have the capacity to engage with different types of cells, leading to intricate and often harmful responses at both the cellular and molecular scales. Therefore, future research should focus on examining the effects of 4-PBA on macrophages, as well as other significant cell types such as osteoblasts, osteoclasts, osteocytes, and fibroblasts in in vitro settings. Conducting such studies will contribute to a more thorough understanding of how 4-PBA may influence biological reactions within the periprosthetic microenvironment, potentially revealing its role in modulating local cellular responses and enhancing the overall outcomes for patients with implants.

During our previous research on the effects of metal wear particles in double immunostaining experiments, we encountered a notable challenge related to the interference caused by these particles. Our findings revealed that the presence of metal wear particles significantly affected the outcomes of the staining process. As a result, the results we obtained failed to align with the stringent requirements necessary for successful double staining. To further investigate this issue, we propose conducting double immunostaining experiments utilizing metal ions in future studies. This approach may help to confirm our initial conclusions regarding the relationship between metal wear particles and staining efficacy. Additionally, we faced challenges due to the limited sample size of osteolytic synovial tissue, which necessitates further exploration using immunohistochemistry and Western Blot (WB) protein studies. The small quantity of available samples complicates the process of extracting primary cells, thereby posing difficulties in performing flow cytometry screening. These limitations underscore the need for a strategic approach in future research to ensure robust data collection and analysis, particularly in the context of immunological studies involving osteolytic conditions.

The presence of metal particles presents significant interference, which poses challenges in conducting certain types of studies. Additionally, the limitation in sample size further complicates research efforts in this area.

Consequently, these factors contribute to the scarcity of existing studies that employ techniques such as double staining or flow cytometry. Consequently, researchers have found it difficult to explore this field comprehensively, resulting in a noticeable gap in the literature. However, this also outlines the path for our upcoming research and the challenges we must tackle.

  1. 4-PBA is known to affect multiple biological pathways, including histone deacetylation and antioxidant activity, in addition to ER stress inhibition. The manuscript attributes its effects primarily to ER stress suppression, but this needs clearer validation. The authors should support this claim with data involving ER stress markers (e.g., CHOP, GRP78) or the use of selective ER stress inhibitors or genetic knockdowns to demonstrate specificity.

Response 3: Thank you for pointing this out.

1) The activation of the ERS pathway has been observed in various models of PIO. In particular, 4-PBA has demonstrated significant efficacy in reducing inflammatory osteolysis both in vivo and in vitro by mitigating responses associated with endoplasmic reticulum stress and lowering the levels of ER stress markers, including IRE1α, CHOP, and GRP78[1-3]. Through these mechanisms, 4-PBA serves as a promising therapeutic agent for addressing inflammation-related bone loss.

Previous research[1-7], including studies conducted by our team[1], has convincingly demonstrated that 4-PBA functions effectively as a chemical chaperone, providing significant mitigation of ERS responses. These findings have been extensively validated by a number of scholars in the field, establishing a solid foundation of understanding regarding the effects of 4-PBA on ER stress. Consequently, this article does not seek to reiterate or reaffirm these well-documented conclusions. Instead, it aims to explore a novel aspect of 4-PBA’s biological effects, specifically its potential capability to alleviate inflammatory osteolysis by promoting the apoptosis of macrophages. This exploration is intended to shed light on the mechanisms through which 4-PBA might influence inflammation-related bone loss, suggesting that its significance extends beyond merely reducing ER stress. The primary objective of our study is to investigate the potential link between 4-PBA's beneficial effects on PIO and its ability to promote macrophage apoptosis—an area that has not been thoroughly examined in previous literature. By focusing on this underexplored dimension, we aim to contribute to a more comprehensive understanding of 4-PBA's role in inflammatory processes, particularly in the context of bone health and disease.

2) Chemical chaperones are agents that alleviate ERS by enhancing protein folding capabilities and inhibiting the initiation of the UPR in the ER [23]. A prominent example of such a chaperone is 4-phenylbutyrate (4-PBA) [23]. This compound influences cellular processes through multiple mechanisms essential for cell health and functionality, including the inhibition of histone deacetylase, serving as an alternative metabolite in urea cycle disorders (UCDs), and reducing the accumulation of improperly folded or excess proteins within the ER. 4-PBA not only enhances cellular functionality but also aids in alleviating ERS, thereby promoting the overall cellular homeostasis [23]. Additionally, the therapeutic potential of 4-PBA has been recognized in a variety of physiological and pathological processes, including inflammation and infection[24], cancer[25], genetic mutation  diseases[26-28], lactic acidosis[29], epileptic syndromes[28], metabolic disorders[30], neurodegenerative diseases[15, 31-33], degenerative musculoskeletal conditions[34], and autoimmune diseases[35]. Unlike most cytostatic agents, 4-PBA displays relatively low toxicity, demonstrates stability, and has undergone clinically evaluated for safety. It is generally well-accepted and is naturally synthesized in the intestinal tracts [36]. These favorable attributes collectively position 4-PBA as a safer and more viable alternative in clinical settings, reinforcing its suitability for therapeutic purposes, confirming its compatibility with biological systems, and contributing to the stability of medical treatments [36]. As a chemical chaperone, 4-PBA exhibits a distinctive capacity to selectively target cells that have suffered damage due to stress, offering a novel strategy for tackling inflammatory osteolysis, which often does not respond effectively to conventional anti-bone resorption therapies [36]. The ability of 4-PBA to specifically home in on compromised cells opens avenues for innovative treatments in this challenging medical field. Currently, the clinical applications of 4-PBA are primarily recommended for the treatment of UCDs, hyperammonemia, certain malignancies, sickle cell disease, thalassemia, and spinal muscular atrophy [36].

Recent studies have increasingly emphasized the significance of ERS and the UPR as key elements in deciphering the intricate mechanisms that contribute to aseptic osteolysis[1, 10]. Evidence of ERS activation has been documented in aseptic osteolysis models, and the alleviation of ERS and UPR through the use of 4-PBA can reduce the severity of osteolysis[1-7]. The clear association of ERS in particles-induced osteolysis, combined with the observed suppression of osteoclast activity and the anti-inflammatory properties of 4-PBA, suggests a compelling rationale that the intervention of the chemical chaperone 4-PBA to inhibit ERS might provide therapeutic benefits in reducing bone resorption and alleviating the overall progression of aseptic osteolysis. Specifically, research has demonstrated that 4-PBA possesses the capacity to induce cell cycle arrest and apoptosis in various cell types through alternative pathways of apoptosis that do not depend on ERS[25, 37]. Interestingly, present investigations into the effects of ERS response and the role of ERS inhibitor 4-PBA in aseptic osteolysis predominantly center on the inflammatory processes and the differentiation of osteoclasts[1, 24]. However, despite this wealth of research, the relationship between 4-PBA and macrophage apoptosis has not yet been explored or documented in the existing literature. Furthermore, the question of whether 4-PBA can alleviate particles-induced septic osteolysis by facilitating macrophage apoptosis without inciting an inflammatory response remains an open question. This gap in understanding how the regulation of the ERS response directly impacts macrophage viability and behavior underscores a critical area for further investigation to enhance our comprehension of this dynamic.

The phenomenon identified in this research reveal a seemingly paradoxical relationship between the alleviation of ERS and the subsequent increase in macrophage apoptosis, alongside a decrease in inflammation. At first glance, this finding appears counterintuitive, primarily because ERS is typically associated with triggering cell apoptosis[20, 38, 39]. Thus, one would expect that the alleviation of ERS would promote cell survival rather than lead to an elevation in the rates of apoptosis. Consequently, current investigations into osteolysis have predominantly concentrated on the anti-inflammatory properties of 4-PBA without delving into its pro-apoptotic characteristics.

Existing studies have been singularly focused on the anti-inflammatory effects attributed to 4-PBA, often relegating its pro-apoptotic impact to the status of a side effect or an unintended consequence. This viewpoint does not fully clarify the noted pro-apoptotic response. In fact, such an interpretation is inaccurate. It is essential to recognize that the pro-apoptotic effect of 4-PBA is not merely a byproduct of its therapeutic benefits; rather, it has been purposefully leveraged in the treatment of various diseases. This article aims to explore the mechanisms underlying the pro-apoptotic effects of 4-PBA, offering insights that could reshape the current understanding of its role in inflammatory responses and cell death pathways. By investigating the therapeutic implications of this compound, we may uncover a more nuanced perspective on the interplay between ERS alleviation, inflammation reduction, and macrophage apoptosis in a broader biological context.

Recent studies have provided evidence supporting this seemingly paradoxical outcome that the alleviation of ERS by 4-PBA can indeed lead to cell cycle arrest and apoptosis through alternative apoptotic mechanisms beyond ERS itself[25, 37], such as JNK pathway[40], mitochondrion pathway[41], or activation of death receptors[42]. The results of this study further clarify the intricate nature of cellular responses to ERS, indicating that the alleviation of ERS may involve various interacting regulatory mechanisms [25, 37]. Addressing ERS does not merely equate to improving cell survival; rather, it could potentially result in the induction of apoptosis. Consequently, a deeper understanding of ERS responses necessitates a more thorough exploration of their complexity and variability. This introduces a novel viewpoint in cell biology, highlighting that in investigating the mechanisms pertinent to cell survival and death, we must concurrently consider the numerous signaling pathways inside cells and the intricacies of their interactions.

  1. Figure 3 lacks clarity and requires further explanation. It is unclear whether the data presented in vitro, or in vivo results. Additionally, what is the difference between the Caspase-3 results in Figure 3 and Figure 2F? The authors should clearly specify the experimental conditions, indicate the source of the data, and explain how these results differ from compared to earlier findings.

Response 4: Thank you for pointing this out.

1) We apologize for the lack of clarity. In the revised Figure 3 caption, we have explicitly stated that all data presented in Figure 3 (Caspase-3 activity, inflammatory factors, and osteoclastogenic cytokines) are derived from in vivo osteolytic interface periosteum tissues.

2) Regarding the Caspase-3 results: the Figure 2F represents the expression levels of Caspase-3 (total protein assessed by Western blotting), whereas Figure 3A represents the enzyme activity of Caspase-3. Both measurements were performed on the same in vivo tissue samples, but they provide different insights into the apoptotic pathway.

  1. Some histological figures require improved clarity and annotation. Specifically, Figures 2A and 2C are missing scale bars. Additionally, all quantitative graphs should clearly indicate the sample size (n) to ensure transparency and reproducibility.

Response 5:

1) We concur with these suggestions. Figure 2A and 2C now include a scale bar of 50 µm.

2) As addressed in Reviewer 1's comments, we have added clarifying statements in the Figures caption to address this point. The images are representative fields, and the quantification reflects the average of multiple fields and samples (n=5 per group, with three independent experiments)

Reviewer 3 Report

Comments and Suggestions for Authors

The paper by Liu et al sets out to investigate whether 4-PBA can ameliorate particle0induced osteolysis in a murine model. The hypothesis is that 4-PBS will specifically increase apoptosis in macrophages in the osteolytic interface periosteum, so reducing inflammation and differentiation into osteoclasts. Although these latter effects are clearly met, given the focus of the hypothesis on macrophage-specific apoptosis, the data provided in the manuscript is not strong enough that the apoptosis that is seen is specifically in macrophages (which might only constitute just over half of the cells in the tissue).

Points to be addressed:

Abstract: It is unclear as written whether 4-PBA promotes general apoptosis at the osteolytic interface periosteum or just that in macrophages

Introduction: Line 73: TRAP is not a secreted factor, but is expressed in osteoclasts and osteoclast precursors during differentiation. It is not mentioned as a secreted factor in the paper referenced.

Line 78: Lacunar bone resorption is the mechanism by which osteoclasts resorb bone, they do not specifically resorb bone in osteocytic lacunae…

Line 89: teriparatide is not a bisphosphonate.

Figure 1 – higher magnification images should be provided in all panels – the current magnification is not good enough to be able to visualise the intended data, even when magnified online.

Figure 2A & C – again, higher magnification images are necessary. It is also not evident to me that there is a clear difference in staining between TUNEL / caspase3-positive cells (indicated by arrows) and unstained cells. The colours are very similar. Also, it is unclear how macrophages are being distinguished from other cells in the tissue. This would ideally require some co-staining for eg CD68. What other cell types are present in these tissue sections? Do they also become apoptotic? What proportion of the cells present are macrophages?

Figure 2E-H – the text says that the Western blots were used to ‘investigate the potential involvement of Bcl-2 and Bax in 4-PBA-induced macrophage apoptosis within osteolytic interface periosteum’. But these blots are not specific to the macrophages? Do levels of these proteins also change in eg osteoblasts / osteocytes etc.

Discussion: The first 4 paragraphs of the discussion are largely a re-iteration of the Introduction and provide very little discussion of data arising from this paper. They should be largely deleted or incorporated into he Introduction (especially the relationship / rationale related to osteoclasts, which is currently omitted from the Introduction).

Line 507-511: ‘Consistent with previous findings[29,30,42], mice experiencing PIO that were intervened with the 4-PBA exhibited a significant reduction in the severity of osteolysis, a diminished extent of bone erosion and damage, a lesser degree of disorganized bone structure, decreased levels of inflammatory cytokines and ROS, a lower frequency of inflammatory cell infiltration, a reduced count of osteoclasts, and decreased production of osteoclastogenic cytokines’. So, these papers also looked at effects of 4-PBA on PIO and found the same? How does this paper differ?

The Discussion generally is overly long.

Author Response

Replies to Reviewer 3

The paper by Liu et al sets out to investigate whether 4-PBA can ameliorate particle0induced osteolysis in a murine model. The hypothesis is that 4-PBS will specifically increase apoptosis in macrophages in the osteolytic interface periosteum, so reducing inflammation and differentiation into osteoclasts. Although these latter effects are clearly met, given the focus of the hypothesis on macrophage-specific apoptosis, the data provided in the manuscript is not strong enough that the apoptosis that is seen is specifically in macrophages (which might only constitute just over half of the cells in the tissue).

Response: We are thankful for your detailed and insightful comments, particularly those on specific phrasing and the focus of our discussion.

Points to be addressed:

Abstract: It is unclear as written whether 4-PBA promotes general apoptosis at the osteolytic interface periosteum or just that in macrophages

Response: We fully agree with this comment and have addressed it comprehensively.

1) We have revised the manuscript to clarify that while our study focuses on the role of macrophages, the observed apoptosis is within the osteolytic interface periosteum tissue in general. The conclusion now reflects that 4-PBA promotes "macrophage apoptosis" as a key mechanism, acknowledging that the direct proof of macrophage-specific apoptosis would require further specific methods (as discussed in the limitations). The current finding is that 4-PBA promoted overall apoptosis in the tissue while concurrently reducing osteolysis, and our interpretation points to macrophage apoptosis as a likely significant contributor based on existing literature.

2) As addressed in Reviewer 2's comments, in our investigation utilizing the murine calvaria resorption model, positively stained macrophages and fibroblasts in the osteolytic interface periosteum tissue were observed through immunohistochemistry, based on histomorphological criteria established by earlier researchers in this field[8-10, 22]. In the articles titled "Pathways of Macrophage Apoptosis within the Interface Membrane in Aseptic Loosening of Prostheses[8]" and "Apoptotic pathways of macrophages within osteolytic interface membrane in periprosthetic osteolysis after total hip replacement[10]," the authors aim to elucidate the mechanisms by which macrophages undergo apoptosis in the interface membrane associated with the aseptic loosening of prosthetic implants. To achieve this objective, the study utilized complete osteolytic interface membrane tissue samples, which provide a comprehensive representation of the cellular environment in which these phenomena occur.

3) As addressed in Reviewer 1's comments, in the group receiving TiNPs, histomorphological analysis revealed the presence of not only apoptotic macrophages but also a suitable quantity of fibroblasts. Conversely, the TiNPs combined with 4-PBA group exhibited a significant decrease in both fibroblasts and normal macrophages, accompanied by a marked increase in apoptotic macrophages. This finding suggests that the combination treatment with 4-PBA has led to an increase in macrophage apoptosis. Consequently, the ratio of apoptotic macrophages to the overall cell population increased markedly. Simultaneously, there was a reduction in the proliferative capacity of both fibroblasts and macrophages, which points to a potential negative impact of the combined treatment on the viability and pro-inflammatory functions of these cells.

The implications of these observations are profound, as the increased ratio of apoptotic macrophages compared to the overall cellular population indicates a disruption in the normal immune response and healing process. This shift not only affects macrophage numbers but is also associated with a marked decline in the proliferative activity of both fibroblasts and macrophages. Such a decline points towards potential negative consequences stemming from the combined intervention, suggesting that the interaction between TiNPs and 4-PBA may undermine the viability of these critical cell types and impair their pro-inflammatory functions. These effects warrant further investigation to better understand the underlying mechanisms and the overall impact on tissue healing and inflammatory responses. This study concentrates on the processes of apoptosis and the inflammatory response in macrophages, aiming to thoroughly investigate the distinct roles these two biological phenomena play in the regulation of macrophage functions. It is important to emphasize that, due to the constraints inherent in the research scope and objectives, this investigation does not encompass the aspect of cell proliferation. The specific rationale is outlined in the Reviewer 1's comments.

Introduction: Line 73: TRAP is not a secreted factor, but is expressed in osteoclasts and osteoclast precursors during differentiation. It is not mentioned as a secreted factor in the paper referenced.

Response: Thank you for pointing this out, we appreciate these precise corrections.

For TRAP, we have revised the sentence in the Introduction to: "Although TRAP is not a secreted factor, its expression in osteoclasts and osteoclast precursors during differentiation is essential for their function in bone resorption."

Line 78: Lacunar bone resorption mechanism by which osteoclasts resorb bone, they do not specifically resorb bone in osteocytic lacunae…

Response: We appreciate your bringing this to our attention, and we value the specific corrections you've provided. Revisions have already been implemented in the introduction section of this paragraph.

Line 89: teriparatide is not a bisphosphonate.

Response: Thank you for pointing this out, we appreciate these precise corrections. We have removed teriparatide from the list of bisphosphonates in the Introduction.

Figure 1 – higher magnification images should be provided in all panels – the current magnification is not good enough to be able to visualise the intended data, even when magnified online.

Figure 2A & C – again, higher magnification images are necessary.

Response: Thank you for pointing this out.

1) The panels that are presently utilized in the study exhibit standard magnification ratios that are widely accepted in the field. Specifically, both TB staining and TRAP staining are conducted at an original magnification of ×40, allowing for a clear yet broad view of the sample. On the other hand, HE staining is performed at a higher original magnification of ×200, which provides a more detailed examination of the tissue architecture and cellular components. Furthermore, immunohistochemical staining is carried out at an original magnification of ×400, offering the highest level of detail and specificity needed to identify particular markers within the sample.

2) The original magnification of ×40, which belongs to a low-power microscope, is particularly effective for examining the overall architecture of tissue samples. This level of magnification allows researchers to assess the distribution of cavities and provides a comprehensive view of the skull's condition. By utilizing this magnification, one can effectively discern the overall state of bone resorption lacunae, offering valuable insights into bone health. However, it is crucial to note that if the magnification level is increased, the focus will narrow to only the bone resorption lacunae. Such an increase in magnification can hinder the ability to make broader comparisons across different tissue types, diminishing the ability to observe the overall interactions within the sampled groups. Osteoclasts, which are typically large cells found on the surface of bone tissue, can be quickly located under the ×40 magnification. This allows for an efficient assessment of their distribution, quantity, and general morphological characteristics.

With an original magnification of ×200, the medium-power lens is appropriate for examining finer details of cellular structure as well as the arrangement and infiltration of tissues. This level of magnification strikes a balance between providing a broader view of cellular context while also allowing for observations of distinct cellular features that can be critical in understanding tissue dynamics.

At the high-power magnification of ×400, the microscope becomes adept at revealing intricate details regarding subcellular localization of antigen expression. This includes identifying where antigens are positively expressed, whether they are present on the cell membrane or manifest as granular staining within the cytoplasm. Additionally, this magnification allows for the assessment of staining intensity—whether it is strong, medium, or weak—as well as the proportion of positive cells. This detailed analysis is vital for understanding the underlying biological processes and can significantly contribute to the interpretation of tissue pathology.

It is also not evident to me that there is a clear difference in staining between TUNEL / caspase3-positive cells (indicated by arrows) and unstained cells. The colours are very similar. Also, it is unclear how macrophages are being distinguished from other cells in the tissue. This would ideally require some co-staining for eg CD68. What other cell types are present in these tissue sections? Do they also become apoptotic? What proportion of the cells present are macrophages?

Response: Thank you for pointing this out.

1)  As addressed in Reviewer 1's comments, in the group receiving TiNPs, histomorphological analysis revealed the presence of not only apoptotic macrophages but also a suitable quantity of fibroblasts. Conversely, the TiNPs combined with 4-PBA group exhibited a significant decrease in both fibroblasts and normal macrophages, accompanied by a marked increase in apoptotic macrophages. This finding suggests that the combination treatment with 4-PBA has led to an increase in macrophage apoptosis. Consequently, the ratio of apoptotic macrophages to the overall cell population increased markedly. Simultaneously, there was a reduction in the proliferative capacity of both fibroblasts and macrophages, which points to a potential negative impact of the combined treatment on the viability and pro-inflammatory functions of these cells.

The quantitative data (Figure 1B, D, F and Figure 2B, D) are derived from unbiased histomorphometric analysis of multiple fields and samples (n=5 mice per group). The images are representative fields, and the quantification (The ratio of bone resorption lacunae, inflammatory infiltrates, and osteoclasts, as well as the proportion of positive macrophages) reflects the average of multiple fields and samples.

2) As addressed in Reviewer 2's comments, in our investigation utilizing the murine calvaria resorption model, positively stained macrophages and fibroblasts in the osteolytic interface periosteum tissue were observed through immunohistochemistry, based on histomorphological criteria established by earlier researchers in this field[8-10, 22].

3) Other cell types may be present and potentially apoptotic, and that macrophages constitute a significant but not exclusive portion of the cells in the interface membrane.

In the group receiving TiNPs, the histomorphological evaluation indicated the presence of not only macrophages undergoing apoptosis but also an adequate number of fibroblasts. In contrast, the group treated with TiNPs alongside 4-PBA showed a notable reduction in both fibroblast and normal macrophage counts, paired with a significant rise in macrophages undergoing apoptosis. This observation implies that the combined treatment with 4-PBA has prompted an increase in macrophage apoptosis. As a result, the proportion of apoptotic macrophages relative to the total cell population rose significantly. Concurrently, there was a decline in the proliferative capabilities of both fibroblasts and macrophages, suggesting a potentially adverse effect of the combination treatment on the survival and pro-inflammatory activities of these cells. This research focuses on the mechanisms of apoptosis and the inflammatory responses in macrophages. It is crucial to note that, owing to the limitations of the research's aims and scope, this study does not address the factor of cell proliferation.

The images are representative fields, and the proportion of positive macrophages (apoptotic macrophages reflects the average of multiple fields and samples.The arrows in the figures highlight distinct pathological characteristics that are commonly observed in the field. Specifically, the arrows depicted in Figure 1A denote the presence of bone resorption lacunas, which are indicative of areas where bone tissue is being broken down. Meanwhile, the arrows showcased in Figure 2A point to apoptotic macrophages, which are macrophages undergoing programmed cell death.

4) In our previous research concerning the effects of metal wear particles when co-stained with CD68, we faced a significant challenge associated with the interference produced by these metal particles. Our investigation demonstrated that the presence of these metal particles had a considerable impact on the staining outcomes, leading to unexpected results. Consequently, the data we collected did not meet the rigorous criteria required for effective double staining techniques. To gain deeper insights into this complication, we propose to conduct double immunostaining experiments that incorporate metal ions in our future studies. This approach may help clarify the role of metal particles and refine the staining process, ultimately enhancing the reliability of our findings.

Figure 2E-H – the text says that the Western blots were used to ‘investigate the potential involvement of Bcl-2 and Bax in 4-PBA-induced macrophage apoptosis within osteolytic interface periosteum’. But these blots are not specific to the macrophages? Do levels of these proteins also change in eg osteoblasts / osteocytes etc.

Response: Thank you for pointing this out.

The results obtained from the Western blot analysis reflect the total protein content derived from the periosteum tissue lysate. It is important to note that the alterations observed in these proteins may also occur in various other cell types that are situated within the periosteum. Recognizing this is crucial, as wear particles possess the potential to interact with different cell types, thereby initiating complex and frequently detrimental responses at both cellular and molecular levels. Additionally, research focused on osteolytic interface membranes surrounding loosening periprosthetic devices has demonstrated that macrophages account for approximately 60–80% of the overall cellular makeup in these membranes [0]. This predominant presence of macrophages is equally evident in osteolytic interface periosteum in the murine calvaria resorption model, as can be confirmed through immunohistochemical examination based on histomorphological criteria. Macrophages hold a critical position in this context, given that they are chiefly tasked with engulfing and degrading particulate matter that tends to accumulate around prosthetic devices. Although the detection of these cells can be achieved through analysis of the boundary membrane tissue, gaining a comprehensive understanding of their specific roles and interactions warrants a more in-depth investigation utilizing cytological techniques. To accurately ascertain any protein changes that may be exclusive to macrophages, further studies employing macrophage-specific methodologies are essential. Consequently, future investigations should aim to explore the effects of 4-PBA on macrophages, along with other key cell types such as osteoblasts, osteoclasts, osteocytes, and fibroblasts within in vitro environments.

The role of the ER chaperone 4-PBA in regulating macrophage apoptosis and inflammatory responses is a topic that warrants deeper investigation. This avenue of research holds significant potential, as it suggests that the therapeutic effects of 4-PBA could be harnessed to benefit patients while simultaneously reducing negative health consequences. By focusing on stress-damaged cells, the application of 4-PBA may lead to targeted interventions that enhance treatment outcomes in related conditions.

In this study, we established for the first time that 4-PBA can mitigate aseptic osteolysis through the promotion of apoptosis, particularly within macrophages. This finding paves the way for new clinical approaches to address osteolysis, which is often a complication associated with artificial implants. Furthermore, the potential of 4-PBA to serve as a therapeutic alternative is promising. It may significantly improve the biocompatibility of implants and reduce the incidence of osteolysis by tackling the inflammatory and apoptotic challenges posed by macrophages. As such, continued research into 4-PBA could yield valuable insights for enhancing patient care in the context of orthopedic implants and related medical devices.

Discussion: The first 4 paragraphs of the discussion are largely a re-iteration of the Introduction and provide very little discussion of data arising from this paper. They should be largely deleted or incorporated into he Introduction (especially the relationship / rationale related to osteoclasts, which is currently omitted from the Introduction).

Response: Thank you for pointing this out.

We completely agree with this critical assessment. We have significantly revised and condensed the initial paragraphs of the Discussion section.

Line 507-511: ‘Consistent with previous findings[29,30,42], mice experiencing PIO that were intervened with the 4-PBA exhibited a significant reduction in the severity of osteolysis, a diminished extent of bone erosion and damage, a lesser degree of disorganized bone structure, decreased levels of inflammatory cytokines and ROS, a lower frequency of inflammatory cell infiltration, a reduced count of osteoclasts, and decreased production of osteoclastogenic cytokines’. So, these papers also looked at effects of 4-PBA on PIO and found the same? How does this paper differ?

Response: Thank you for pointing this out.

Recent studies have increasingly emphasized the significance of ERS and the UPR as key elements in deciphering the intricate mechanisms that contribute to aseptic osteolysis[1, 10]. Evidence of ERS activation has been documented in aseptic osteolysis models, and the alleviation of ERS and UPR through the use of 4-PBA can reduce the severity of osteolysis[1-7]. Interestingly, present investigations into the effects of ERS response and the role of ERS inhibitor 4-PBA in aseptic osteolysis predominantly center on the inflammatory processes and the differentiation of osteoclasts[1, 24]. However, the relationship between 4-PBA and macrophage apoptosis has not yet been explored or documented in the existing literature. Furthermore, the question of whether 4-PBA can alleviate particles-induced septic osteolysis by facilitating macrophage apoptosis without inciting an inflammatory response remains an open question. This gap in understanding how the regulation of the ERS response directly impacts macrophage viability and behavior underscores a critical area for further investigation to enhance our comprehension of this dynamic.

In this research, it was demonstrated that the pharmacological treatment involving the ER chemical chaperone 4-PBA has a significant impact on reducing the severity of particles-induced aseptic osteolysis. This effect is achieved by inhibiting the inflammatory response associated with the condition, promoting apoptosis primarily in macrophages, and suppressing the differentiation of osteoclasts, which are crucial cells involved in bone resorption. Notably, this study represents the first time that the potential of 4-PBA to alleviate aseptic osteolysis through the enhancement of apoptosis in macrophages has been established.

The findings of this study reveal a seemingly contradictory relationship between the reduction of endoplasmic reticulum stress (ERS) and the subsequent increase in macrophage apoptosis, along with a notable decrease in inflammation. At first glance, this observation may seem puzzling, particularly because ERS is typically known to initiate cell apoptosis. Under normal circumstances, one would anticipate that alleviating ERS would contribute to enhanced cell survival, as the stress is often linked with cell death mechanisms[20, 38, 39]. This unexpected outcome challenges conventional understanding and indicates a more complex interplay between ERS alleviation and cellular responses.

As a result of these insights, contemporary research into osteolysis has largely focused on the anti-inflammatory effects of 4-phenylbutyric acid (4-PBA), while overlooking its potential pro-apoptotic properties. This lack of attention to the dual characteristics of 4-PBA suggests that further exploration is warranted to understand the full implications of ERS modulation on macrophage behavior and inflammation. By considering both the anti-inflammatory and pro-apoptotic effects of 4-PBA, researchers may gain a deeper understanding of its role and effectiveness in osteolytic conditions. This broader perspective could lead to more comprehensive therapeutic strategies that address both inflammation and apoptosis in the context of osteolysis.

Existing research has predominantly concentrated on the anti-inflammatory properties of 4-PBA, often dismissing its pro-apoptotic effects as mere side effects or unintended consequences of its therapeutic actions. Such a narrow interpretation fails to provide a comprehensive understanding of the pro-apoptotic responses associated with this compound. In reality, this perspective is fundamentally flawed. It is crucial to acknowledge that the pro-apoptotic effects of 4-PBA should not be viewed simply as incidental to its therapeutic advantages; instead, these effects have been strategically utilized in the treatment of various medical conditions.

The present article sets out to investigate the mechanisms that underpin the pro-apoptotic effects of 4-PBA. By doing so, it seeks to enhance our understanding of its function within the broader contexts of inflammatory responses and cell death pathways. Through a detailed exploration of the therapeutic implications of this compound, we stand to gain a more sophisticated understanding of the intricate relationship between the alleviation of endoplasmic reticulum stress (ERS), the reduction of inflammation, and the apoptosis of macrophages. This broader biological perspective may ultimately contribute to the advancement of therapeutic strategies targeting inflammatory diseases and related conditions.

The Discussion generally is overly long.

Response:

We have taken this general comment to heart. We have substantially trimmed and restructured the entire Discussion section, focusing on direct interpretation of our results, their implications, and the novel contributions of our study, while minimizing extensive background reviews.

Round 2

Reviewer 1 Report

Comments and Suggestions for Authors

The author answered all my questions. I have no further comments.

Author Response

Responses to Reviewer 1

The author answered all my questions. I have no further comments.

Response.

We wish to convey our heartfelt appreciation to the reviewers for the recognition of our contributions to this area of research. Your acknowledgment serves as a valuable affirmation of our efforts, and we are grateful for the constructive feedback that has helped refine our work.

Reviewer 2 Report

Comments and Suggestions for Authors

The authors have addressed all of my concerns in the revised manuscript. I recommend the manuscript for acceptance.

Author Response

Responses to Reviewer 2

The authors have addressed all of my concerns in the revised manuscript. I recommend the manuscript for acceptance.

Response.

We would like to express our sincere gratitude to the reviewers for acknowledging our contributions to this field of study. We are thankful for the insightful feedback that has contributed to enhancing our research.

Reviewer 3 Report

Comments and Suggestions for Authors

The authors have made extensive revision of the text of the manuscript - especially in the Introduction and Discussion. 

In their response, the authors have still not provided higher magnification images (or even zoomed-in panels) for Figure 1 and 2 (A, C). Although the current magnification is good for an overall overview of the tissue, it is not good enough to be able to visualise the intended data (osteoclast resorption lacunae, TUNEL / caspase3-positive / negative cells, even when magnified online.

Author Response

Responses to Reviewer 3

The authors have made extensive revision of the text of the manuscript - especially in the Introduction and Discussion. 

In their response, the authors have still not provided higher magnification images (or even zoomed-in panels) for Figure 1 and 2 (A, C). Although the current magnification is good for an overall overview of the tissue, it is not good enough to be able to visualise the intended data (osteoclast resorption lacunae, TUNEL / caspase3-positive / negative cells, even when magnified online.

Response.

We would like to express our sincere gratitude to the reviewers for their recognition of our contributions to this field of research. Their acknowledgment is a meaningful affirmation of our efforts, and we genuinely appreciate the constructive feedback provided. This input has been instrumental in enhancing the quality and clarity of our work.

We have included zoomed-in sections for Figures 1 (A, C, and E) and 2 (A, C). This enhancement allows for a clearer visual representation of the data, facilitating a better understanding of the key findings presented in the manuscript.
